# SAFE MULTI-OBJECTIVE REINFORCEMENT LEARNING VIA MULTI-PARTY PARETO NEGOTIATION

## ABSTRACT

Safe multi-objective reinforcement learning (Safe MORL) seeks to optimize performance while satisfying safety constraints. Existing methods face two key challenges: (i) incorporating safety as additional objectives enlarges the objective space, requiring more solutions to uniformly cover the Pareto front and maintain adaptability under changing preferences; (ii) strictly enforcing safety constraints is feasible for single or compatible constraints, but conflicting constraints prevent flexible, preference-aware trade-offs. To address these challenges, we cast Safe MORL within a multi-party negotiation framework that treats safety as an external regulatory perspective, enabling the search for a consensus-based multi-party Pareto-optimal set. We propose a multi-party Pareto negotiation (MPPN) strategy built on NSGA-II, which employs a negotiation threshold $\varepsilon$ to represent the acceptable solution range for each party. During evolutionary search, $\varepsilon$ is dynamically adjusted to maintain a sufficiently large negotiated solution set, progressively steering the population toward the $(\varepsilon_{\text{efficiency}}, \varepsilon_{\text{safety}})$-negotiated common Pareto set. The framework preserves user preferences over conflicting safety constraints without introducing additional objectives and flexibly adapts to emergent scenarios through progressively guided $(\varepsilon_{\text{efficiency}}, \varepsilon_{\text{safety}})$. Experiments on a MuJoCo benchmark show that our approach outperforms state-of-the-art methods in both constrained and unconstrained MORL, as measured by multi-party hypervolume and sparsity metrics, while supporting preference-aware policy selection across stakeholders.

## 1 INTRODUCTION

Multi-objective reinforcement learning (MORL) addresses decision-making problems with multiple, often conflicting objectives (Dulac-Arnold et al., 2021). Since no single policy can be optimal across all objectives simultaneously, existing approaches are typically divided into two categories. Single-policy MORL (Chen et al., 2021; Skalse et al., 2022; Kyriakis & Deshmukh, 2022) reduces the multi-objective problem to a scalar one by applying predefined weights, allowing standard RL algorithms to be used directly. However, this scalarization produces a policy tailored to a fixed preference, limiting adaptability across tasks. Multi-policy MORL (Yang et al., 2019; Chen et al., 2019; Xu et al., 2020; Hayes et al., 2022), on the other hand, aims to approximate the Pareto front (PF) by learning a set of non-dominated policies, thereby supporting diverse objective preferences and enabling flexible policy selection in practice.

While MORL has shown promising progress, incorporating safety considerations introduces new challenges. Safe MORL (Huang et al., 2022) aims to optimize multiple objectives while ensuring that agents adhere to safety requirements, thereby preventing hazardous behaviors during training and deployment. One natural formulation is to treat safety as an additional objective alongside performance goals. This enables explicit exploration of safety–performance trade-offs but increases the dimensionality of the objective space, leading to an exponential growth of the Pareto set and making full coverage intractable. Alternatively, safety can be enforced as a hard constraint, giving rise to constrained MORL (CMORL) (Huang et al., 2022; Lin et al., 2024; Gu et al., 2025), which restricts the search to policies that satisfy predefined safety conditions. This avoids dimensionality explosion and directly guarantees safe behavior, but struggles with conflicting or overly strict constraints and lacks adaptability in dynamic environments.

In practice, however, safety is not always absolute. Real-world decision making often requires negotiating between efficiency and safety, where tolerating minor violations of certain safety constraints may yield significant performance gains. For example, as illustrated in Figure 1, cargo-handling robots aim to maximize movement speed and payload capacity while maintaining body stability and limiting energy consumption. Allowing slight violations in stability or energy usage can enable faster transport of larger loads, which may be desirable in time-sensitive scenarios. Such flexibility is difficult to achieve with CMORL, since hard constraints restrict the feasible solution set and often eliminate practically useful trade-offs. Likewise, objective-based MORL approaches may struggle to capture balanced solutions between efficiency and safety when the inclusion of multiple safety objectives causes the Pareto set to expand excessively.

To address these limitations, we reconceptualize Safe MORL as a multi-party negotiation problem, where the safety objectives and efficiency objectives are treated as separate multi-objective decision parties rather than as additional objectives in a single objective space. This formulation enables the search for a common Pareto set that balances efficiency and safety while resolving potential conflicts among safety constraints. Building on this idea, we develop a negotiation-driven evolutionary framework, MPPN-MORL, which integrates multi-party Pareto negotiation into policy search without increasing the dimensionality of the objective space. Our approach flexibly adapts to user-specified preferences over both performance and safety, preserves diversity in the

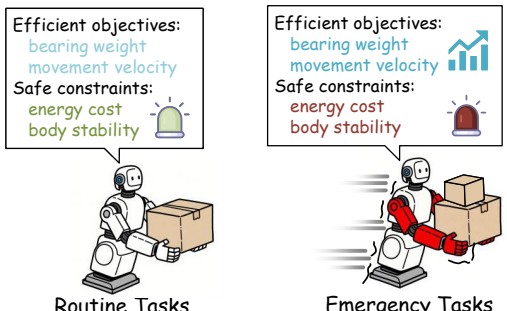

Figure 1: Trade-off between efficiency and safety. Safety constraints are relaxed in emergency tasks to prioritize efficiency objectives.

solution set, and promotes fairness across parties. Extensive experiments on a multi-objective MuJoCo benchmark demonstrate that MPPN-MORL achieves superior trade-offs between efficiency and safety compared to existing MORL and CMORL methods, while effectively handling conflicting safety constraints and supporting preference-aware policy deployment.

## 2 PRELIMINARIES

### 2.1 MULTI-OBJECTIVE DECISION-MAKING

A multi-objective decision-making (MODM) problem involves optimizing multiple, potentially conflicting objectives. Formally, it can be formulated as

$$\min_{\pi_\theta} \mathbf{F}_{\pi_\theta} = \min_{\pi_\theta} \left[ f_1(\pi_\theta), \ldots, f_m(\pi_\theta) \right], \tag{1}$$

where $\pi_\theta$ denotes a parameterized policy, and $f_i(\pi_\theta)$ is the expected performance of $\pi_\theta$ with respect to the $i$-th objective, for $i = 1, \ldots, m$. Unlike single-objective settings, which seek a unique optimal policy, MODM problems typically yield a set of Pareto-optimal solutions, each reflecting a different trade-off among objectives.

### 2.2 CONSTRAINED MULTI-OBJECTIVE REINFORCEMENT LEARNING

Safe MORL extends MODM by enforcing safety constraints, restricting the set of admissible policies. Constrained MORL (CMORL) formalizes this idea using explicit cost functions. Specifically, CMORL introduces $p$ additional cost functions $c_{m+1}, \ldots, c_{m+p}$, each mapping a state-action pair $(s, a)$ to a scalar cost. For a policy $\pi$, the expected cumulative cost under the $(m + i)$-th function is denoted as $c_{m+i}(\pi)$, which must satisfy

$$c_{m+i}(\pi) \leq d_i, \quad \forall i = 1, \ldots, p, \tag{2}$$

where $d_i$ is a predefined safety threshold. The objective in CMORL is to optimize the vector-valued function $\mathbf{F}(\pi)$ representing performance across $m$ objectives, while ensuring that $\pi$ lies within the safe policy set:

$$\Pi_{\text{safe}} = \left\{ \pi \in \Pi \mid c_i(\pi) \leq d_i, \forall i = m + 1, \ldots, m + p \right\}, \tag{3}$$

from which the agent identifies a set of Pareto-optimal policies.

Gu et al. (2025) extend the Pareto frontier concept to safety-constrained MDPs. A policy $\pi \in \Pi_{\text{safe}}$ is safe Pareto-optimal if no other policy in $\Pi_{\text{safe}}$ strictly improves all objectives without violating any constraint. Thus, the central goal of CMORL is to efficiently find such policies, balancing objective performance with constraint satisfaction.

## 3 METHOD

In this section, we introduce the modeling approach for MPMORL and present the MPPN-MORL algorithm, which is capable of selecting appropriate Pareto-optimal policy sets based on the preferences of multiple parties. We first describe the modeling framework for MPMORL, and then elaborate on MPPN-MORL from two key aspects. The detail of MPPN-MORL is shown in Algorithm 2.

### 3.1 MULTI-PARTY MULTI-OBJECTIVE DECISION-MAKING

Multi-party multi-objective decision-making (MPMODM) models scenarios with multiple decision-makers (DMs), where each DM optimizes its own set of objectives and at least one DM faces multiple, potentially conflicting goals. Such scenarios arise naturally in decentralized systems, multi-departmental planning, and cooperative multi-agent environments where each party holds distinct priorities.

In sequential decision-making under uncertainty, MPMODM can be formulated as a multi-party multi-objective Markov decision process (MPMOMDP). In this work, we focus on two parties: the *safety side* and the *efficiency side*. Formally, the problem is defined by the tuple

$$\mathcal{M} = \langle \mathcal{S}, \mathcal{A}, T, \mu, \Gamma, \mathcal{R} \rangle \tag{4}$$

where $S$ is the state space, $\mathcal{A}$ is the action space, $T(s' \mid s, a)$ denotes the state transition probability, and $\mu$ is the initial state distribution. The discount factors are represented by $\Gamma = \gamma^1, \gamma^2$, where $\gamma^k = [\gamma_1^k, \ldots, \gamma_m^k] \in [0,1]^m$ denotes the discount vector for party $k$. The reward function is defined as

$$\mathcal{R}(s, a, s') = \left[ R^1(s, a, s'), R^2(s, a, s') \right], \tag{5}$$

with $R^k(s, a, s') = [r_1^k(s, a, s'), \ldots, r_m^k(s, a, s')]^\top$ representing the $m$-dimensional reward vector for party $k$.

A policy $\pi_\theta : \mathcal{S} \to \mathcal{A}$ guides the agent's behavior. For each party $k \in \{1, 2\}$, its performance is evaluated using a vector of expected discounted returns:

$$J_{i,\pi_\theta}^k = \mathbf{E}_{\tau \sim \pi_\theta} \left[ \sum_{t=0}^\infty (\gamma_i^k)^t r_i^k(s_t, a_t, s_{t+1}) \right], \quad i = 1, \ldots, m. \tag{6}$$

Let $\mathbf{J}_{\pi_\theta}^k = [J_{1,\pi_\theta}^k, \ldots, J_{m,\pi_\theta}^k]^\top$ be the $m$-dimensional return vector for party $k$. The goal of MP-MORL is to identify a set of policies $\{\pi_\theta\}$ that approximates the joint two-party Pareto front, which balances trade-offs between the safety side and the efficiency side.

**Definition 3.1** (One-Party Pareto Dominance). Given two solutions $X, Y \in \mathcal{X}$ and the objective set $F_k$ of the $k$-th DM, $X$ is said to *Pareto dominate* $Y$ with respect to DM $k$, denoted as $X \prec_k Y$, if $f_{k,i}(X) \leq f_{k,i}(Y)$ for all $i \in \{1, \ldots, m_k\}$ and there exists at least one $j$ such that $f_{k,j}(X) < f_{k,j}(Y)$.

**Definition 3.2** (Multi-Party Pareto Dominance). Given two solutions $X, Y \in \mathcal{X}$, $X$ is said to *multi-party Pareto dominate* $Y$, denoted as $X \prec_{\text{MP}} Y$, if $X \prec_k Y$ holds in the local objective space of each DM $k$.

**Definition 3.3** (Multi-Party Pareto Front). Let $\mathcal{X}$ be the solution space. The *multi-party Pareto front* (MP-Pareto front) is defined as the set of solutions that are not multi-party Pareto dominated by any other solution in $\mathcal{X}$, i.e.,

$$\mathcal{PF}_{\text{MP}} = \{X \in \mathcal{X} \mid \nexists Y \in \mathcal{X} \text{ s.t. } Y \prec_{\text{MP}} X\}. \tag{7}$$

In other words, a solution $X$ belongs to the multi-party Pareto front if there does not exist another solution $Y$ that is better than $X$ in the objective spaces of all decision makers simultaneously.

Different from traditional MORL, where Pareto optimality is defined with respect to a centralized objective space, MPMORL introduce a perspective-dependent notion of optimality. A solution regarded as globally Pareto optimal may appear suboptimal from the standpoint of an individual DM with unique preferences. MPMORL requires identifying solutions that not only balance multiple objectives but also ensure diversity and fairness among all participating DMs.

**Example**: Consider a robotic cargo transportation task in which a robot delivers goods from a workstation to a designated target area. In this scenario, two DMs focus on different aspects of the robot's policy: the efficiency party emphasizes transportation speed and payload capacity, whereas the safety party prioritizes energy consumption and body stability.

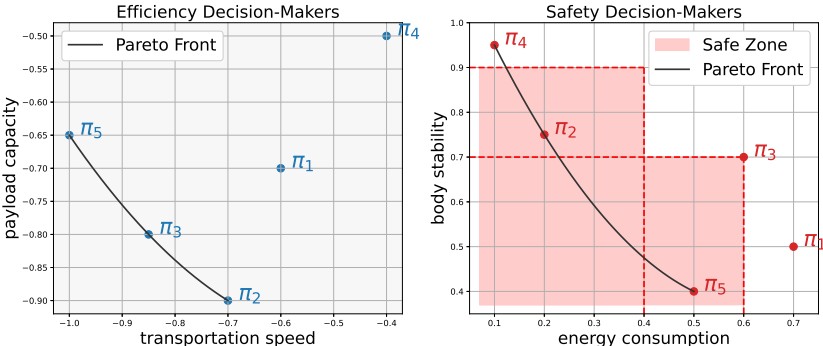

Figure 2: Performance of different policies under MPMORL modeling and CMORL modeling.

As shown in Figure 2 , when modeled as a conventional MORL problem, each policy constitutes a Pareto-optimal solution. However, individual DMs may have diverging preferences: the efficiency party favors polices $\pi_2$, $\pi_3$ and $\pi_5$, while the safety party favors polices $\pi_2$, $\pi_4$, and $\pi_5$. Although all policies are Pareto-optimal in the multi-objective sense, under the multi-party perspective, policies $\pi_1$, $\pi_3$ and $\pi_4$ are dominated by policies $\pi_2$ and $\pi_5$. Therefore, the multi-party Pareto front includes $\pi_2$ and $\pi_5$, eliminating solutions that appear optimal in the centralized view.

When facing conflicting safety constraints, CMORL must perform optimization within a fixed constraint space, thus failing to find all Pareto solutions. While feasible, it lacks the diversity required for negotiation among DMs and cannot effectively resolve conflicts between objectives.We conducted a toy experiment using three representative algorithms in this environment, and the results are presented in the Appendix D.1 .

## 3.2 MULTI-PARTY PARETO NEGOTIATION-BASED NON-DOMINATED SORTING

In MPMODM, the goal is generally to identify a common Pareto set that satisfies all parties. However, such common solutions are often limited. To enlarge the set of negotiable policies while respecting individual preferences, one or both parties may relax their acceptance criteria. We model this process as a bargaining game, where both parties start from an initial compromise level $\varepsilon$ and iteratively negotiate toward their reference thresholds ($\varepsilon_{\text{efficiency}}, \varepsilon_{\text{safety}}$). During this negotiation, the framework ensures that solutions maintain both high quality and uniform coverage, providing a balanced compromise between efficiency and safety.

To address multi-objective decision-making in multi-party scenarios, we propose a *multi-party Pareto negotiation* (MPPN) framework that extends the classical Pareto concept. The detailed procedure of this algorithm is presented in Algorithm 1 . The key idea is to relax the strict dominance relation by introducing an $\varepsilon$-compromise degree with respect to a shared reference solution, allowing each DM to accept solutions that are not strictly superior but still fall within an acceptable margin of improvement relative to this baseline.

Specifically, for each DM, $\varepsilon$-dominance is evaluated against a predefined reference reward vector $\mathbf{r}_{\text{ref}}$. A candidate solution $\mathbf{r}$ is said to $\varepsilon$-dominate the reference if, within the DM's objective subspace, it satisfies $\mathbf{r} \succeq_{\varepsilon} \mathbf{r}_{\text{ref}}$, meaning that its performance is no worse than $\varepsilon \cdot \mathbf{r}_{\text{ref}}$ across all relevant objectives (accounting for optimization direction via element-wise scaling) and strictly better in at least one. This mechanism prevents excessive rejection of solutions due to minor differences and provides a negotiation margin centered on a common target, enabling more practical and sta-

---

**Algorithm 1** Multi-party $\varepsilon$-dominance Sorting

---

**Input**: Candidate set $\mathcal{C}$ with rewards $\{\mathbf{r}_j\}$, reference solution $\mathbf{r}_{\mathrm{ref}}$, compromise vector $\varepsilon = [\varepsilon_1, \varepsilon_2]$, objective partitions $\{J_1, J_2\}$, joint threshold $\tau$
**Output**: Joint $\varepsilon$-front $\mathcal{F}_{\mathrm{joint}}$, local fronts $\{\mathcal{F}_1, \mathcal{F}_2\}$

1: Initialize $\mathcal{F}_{\mathrm{joint}} \leftarrow \emptyset$, $\mathcal{F}_1 \leftarrow \emptyset$, $\mathcal{F}_2 \leftarrow \emptyset$
2: **for** each solution $\mathbf{r}_j \in \mathcal{C}$ **do**
3:     **if** $\mathbf{r}_j$ $\varepsilon$-dominates $\mathbf{r}_{\mathrm{ref}}$ w.r.t. $J_1$ and $J_2$ **then**
4:         Add $\mathbf{r}_j$ to $\mathcal{F}_{\mathrm{joint}}$
5:     **else if** $\mathbf{r}_j$ $\varepsilon$-dominates $\mathbf{r}_{\mathrm{ref}}$ w.r.t. $J_1$ or $J_2$ **then**
6:         Add $\mathbf{r}_j$ to $\mathcal{F}_1$ or $\mathcal{F}_2$
7:     **end if**
8: **end for**
9: **if** $|\mathcal{F}_{\mathrm{joint}}| \geq \tau$ **then**
10:     Shrink $\varepsilon \leftarrow \max(\varepsilon * \Delta\varepsilon, \varepsilon_{\min})$
11: **end if**
12: **return** $\mathcal{F}_{\mathrm{joint}}, \mathcal{F}_1, \mathcal{F}_2$

---

ble preference modeling under conflicting interests.As shown in Figure 3, by reducing the value of $(\varepsilon_{\mathrm{efficiency}}, \varepsilon_{\mathrm{safety}})$, safety constraints can be appropriately relaxed in exchange for tightening policies toward higher-performance regions.

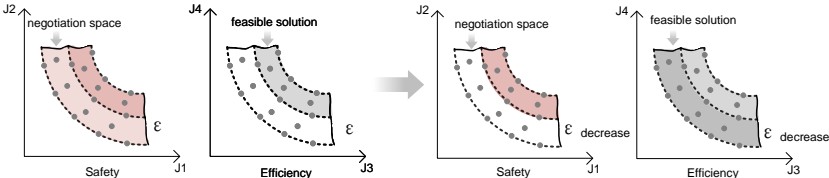

Figure 3: The reward space under $\varepsilon$-dominance varies in accordance with the outcomes of the negotiation process.

To retain appropriate policies during the evolutionary process, we have designed a multi-party non-dominated sorting (MPNDS) method under $\varepsilon$-dominance. This MPNDS approach constructs a ranking dictionary for each DM, which records the Pareto front level of each individual from the perspective of that DM. For $\varepsilon$-dominance, the Pareto level is defined as the minimum policy value that can $\varepsilon$-dominate the reference policy. We then calculate the sum of the levels of each individual across all DMs, and stratify and sort the individuals in ascending order based on the aggregated total level value. This strategy effectively mitigates the dominant impact of overly strict preferences from a single DM on the overall ranking, and better integrates the perspectives of all participating parties. By adopting a level summation mechanism to aggregate multi-party negotiation opinions, MPPN can identify solutions that more fairly reflect the diverse and even potentially conflicting objectives of various stakeholders.

By dynamically tightening $\varepsilon$, the algorithm transitions from broad exploration to focused exploitation. Initially large to allow diverse policies near $\mathbf{r}_{\mathrm{ref}}$, $\varepsilon$ decays only when enough solutions satisfy the condition for all DMs. If one party fails to improve, $\varepsilon$ for the other party is temporarily relaxed to explore better policies.

Overall, the multi-party $\varepsilon$-nondominated sorting first identifies locally preferred solutions within each DM and then integrates them to form a global ranking. This yields a set of $\varepsilon$-nondominated solutions that balance conflicting objectives while maintaining diversity and fairness.

### 3.3 MULTI-PARTY PARETO NEGOTIATION FOR SAFE MORL

To address conflicts among objectives from multiple parties, we propose MPPN-MORL, which incorporates a multi-party negotiation mechanism into evolutionary search. Inspired by NSGA-II(Storn & Price, 1997), the algorithm replaces genetic operators with differential evolution for efficiency and substitutes standard Pareto dominance with an $\varepsilon$-dominance criterion to enable negotiation.

MPPN-MORL initializes a population of candidate policies, evaluates their multi-objective rewards, and assigns party-specific compromise parameters $\varepsilon$ to guide negotiation. In each generation, offspring are generated via differential evolution and combined with parents. Solutions are compared against a reference under the negotiation-based dominance criterion. Jointly dominant solutions tighten $\varepsilon$ to enforce stricter optimality, while if none exist, each party updates its own dominant set, preserving individual preferences.

Population diversity is maintained by prioritizing $\varepsilon$-dominated solutions and filling remaining slots based on crowding distance. This iterative process continues until termination, yielding a negotiation-based Pareto front that balances cooperation and competition, reflecting both individual preferences and mutual consensus.

## 4 THEORETICAL ANALYSIS

This section provides a unified analysis of how the proposed negotiation mechanism guides the search toward high-quality multi-party Pareto solutions. Our results build on (i) the nesting structure of party-wise acceptable sets and (ii) the contraction induced by shrinking $\varepsilon$. All related theoretical derivations and proofs are provided in Appendix E .

### 4.1 THEORETICAL PROOF OF $\varepsilon$-NEGOTIATION CONVERGENCE

We first briefly outline the convergence properties of the proposed $\varepsilon$-dominance negotiation mechanism with dynamic shrinking. The key idea is that, as the tolerance $\varepsilon$ shrinks, the set of mutually acceptable solutions becomes strictly nested, and the evolutionary search progressively focuses on higher-quality regions. Leveraging a time-scale separation between the population mixing and $\varepsilon$-shrinking steps, we can guarantee that the population converges toward the strictest joint Pareto set.

Formally, let $S(\varepsilon)$ denote the joint $\varepsilon$-acceptable set. Starting from an initial large tolerance $\varepsilon_0$ and iteratively shrinking to $\varepsilon_T$, the nested structure ensures:

$$S(\varepsilon_0) \supseteq S(\varepsilon_1) \supseteq \cdots \supseteq S(\varepsilon_T). \tag{8}$$

Under standard assumptions on the evolutionary algorithm (irreducibility, retention, and sufficient mixing), the population is guided layer by layer into stricter subsets, eventually approximating $S(\varepsilon_T)$ with high probability. A detailed proof of this layered convergence is provided in Appendix E.1 .

### 4.2 HARD SAFETY CONSTRAINTS

For hard safety constraints, we enforce them by fixing $\varepsilon_{safety} = 0$, ensuring that the safety agent's acceptable set does not shrink. A detailed proof is provided in Appendix E.2 .

### 4.3 $\varepsilon$-SHRINKING LEADS TO IMPROVED MULTI-PARTY PARETO SOLUTIONS

The key idea behind the improvement is twofold. First, as the negotiation tolerance $\varepsilon$ shrinks, the joint acceptable set $S(\varepsilon)$ becomes strictly smaller and less complex, reducing the solution-space that the evolutionary algorithm must explore. Second, a smaller, lower-complexity set increases the probability that a fixed-budget algorithm samples representative high-quality solutions in every region of $S(\varepsilon)$. Full technical details and proofs are provided in Appendix E.3 .

## 5 EXPERIMENTS

### 5.1 EVALUATION METRICS

In MORL, the most commonly used evaluation metric is the hypervolume (HV) and Sparsity (SP) (Xu et al., 2020; Basaklar et al., 2023; Hu & Luo, 2024; Liu et al., 2025). To evaluate the overall performance in MPMORL scenarios, we employ the Multi-Party Hypervolume (MPHV) and Multi-Party Sparsity (MPSP). Assume that for each DM, an approximated Pareto front $L_k$ is obtained in an $m_k$-dimensional objective space, where $k \in \{1, \ldots, K\}$ indexes the parties and $M_k$ is the number of solutions in $L_k$. Let $r_k \in \mathbb{R}^{m_k}$ be the reference point for the $k$-th party. The HV for $L_k$ is defined as:

$$HV(L_k) = \delta\big(H(L_k, r_k)\big), \tag{9}$$

where

$$H(L_k, r_k) = \{w \in \mathbb{R}^{m_k} \mid \exists j, \ r_k \preceq w \preceq L_{k,j}\}, \tag{10}$$

$L_{k,j}$ is the $j$-th solution in $L_k$, and $\delta(\cdot)$ denotes the Lebesgue measure in $\mathbb{R}^{m_k}$. The relation $\preceq$ is the *weak Pareto dominance* operator, meaning that for two vectors $a, b \in \mathbb{R}^{m_k}$, $a \preceq b$ holds if and only if $a_i \leq b_i$ for all objectives $i$. HV measures the volume of the region dominated by the approximated Pareto set $L_k$ and bounded by the reference point $r_k$, where a larger HV indicates better convergence and diversity properties of the approximation.

By introducing the negotiation thresholds $(\varepsilon_{\text{efficiency}}, \varepsilon_{\text{safety}})$, MPHV aggregates the HV of all parties with preference weights, reflecting the overall performance of the approximated Pareto sets. Its calculation formula is expressed as follows:

$$MPHV = (1 - \varepsilon_{\text{efficiency}}) \cdot HV(L_{\text{efficiency}}) + (1 - \varepsilon_{\text{safety}}) \cdot HV(L_{\text{safety}}). \tag{11}$$

The SP metric is further introduced to evaluate the distribution of solutions along the approximated Pareto front. Unlike HV, which focuses on convergence and overall coverage of the objective space, SP emphasizes the evenness of solution spacing, reflecting how well the algorithm maintains diversity across objectives. Formally, let $L = \{\mathbf{z}_1, \ldots, \mathbf{z}_M\}$ be the approximated Pareto front in an $m$-dimensional objective space, where $M$ is the number of solutions. For each objective dimension $k \in \{1, \ldots, m\}$, the solutions are sorted in descending order by their $k$-th objective value. The sparsity is then computed as:

$$SP(L) = \frac{1}{M-1} \sum_{k=1}^{m} \sum_{j=1}^{M-1} (z_{j,k} - z_{j+1,k})^2, \tag{12}$$

where $z_{j,k}$ denotes the $k$-th objective value of the $j$-th solution after sorting. A lower SP value indicates that the solutions are more evenly distributed along the Pareto front. Therefore, SP serves as a complementary indicator to HV, as it directly measures the diversity of solutions rather than the dominated volume.

Analogous to MPHV, we extend SP to the multi-party setting by defining the Multi-Party Sparsity (MPSP). Specifically, MPSP aggregates the sparsity values of all parties under negotiation thresholds, capturing the overall evenness of solution distribution across different parties. Its formulation is given as:

$$MPSP = (1 - \varepsilon_{\text{efficiency}}) \cdot SP(L_{\text{efficiency}}) + (1 - \varepsilon_{\text{safety}}) \cdot SP(L_{\text{safety}}). \tag{13}$$

This metric reflects the overall quality of the Pareto approximations across all parties. A higher MPHV indicates that the solutions perform well on average for individual parties, maintaining good convergence and diversity. In contrast, a lower MPSP value signifies that the algorithm achieves a well-spread set of solutions for each party and avoids clustering or large gaps between adjacent solutions.

## 5.2 ENVIRONMENT SETTINGS

Based on the MuJoCo (Todorov et al., 2012) and MO-MuJoCo (Xu et al., 2020) benchmark, we developed a MPMO MuJoCo benchmark to evaluate the performance of the proposed algorithms within the MuJoCo framework. This benchmark consists of six continuous robotic locomotion control tasks: MP-HalfCheetah, MP-Walker, MP-Hopper, MP-Pusher, MP-Swimmer, and MP-Humanoid. Each task involves two decision-making parties, namely the safety party and the efficiency party, where each party is associated with two distinct objectives.

We also conducted tests in discrete environments on the commonly used Fruit Tree Navigation (FTN) benchmark (Yang et al., 2019) with different depths. We divided the six objectives into two parties, where each party optimizes three objectives.

The definitions of objectives and reward formulations for all experimental environments are detailed in Appendix C .

## 5.3 BASELINES

To demonstrate the advantages of the MPMORL formulation and to evaluate the effectiveness of the proposed MPPN-MORL algorithm, we conducted experiments against leading methods from

both domains. For CMORL, we first adopted LP3 (Huang et al., 2022) as a baseline algorithm. We also adopted the state-of-the-art algorithm CR-MOPO (Gu et al., 2025). For MORL, we employed PGMORL (Xu et al., 2020), advanced approach designed for continuous state–action spaces. We also selected MOAC (Zhou et al., 2024) and MOCHA (Hairi et al., 2025), the latest cutting-edge methods in the field of MORL, to conduct comparative experiments. Notably, CR-MOPO-S (Gu et al., 2025), which reformulates the safety constraint in CR-MOPO as an additional objective, can also be viewed as a MORL algorithm.

We also conducted comparative experiments on the FTN environment against the Envelope (Yang et al., 2019) and PD-MORL (Basaklar et al., 2023) algorithms.

Furthermore, to evaluate the effectiveness of the MPPN-MORL algorithm, we performed an ablation study in which the MPPN component is removed, and only the MPNDS (Liu et al., 2020) component is employed during the evolutionary process; this variant is referred to as MPPN-ablated.

Further details regarding the algorithmic procedures and parameter settings of the baseline methods are provided in the supplementary material.

## 5.4 RESULTS

We evaluate the proposed methods on the developed continuous control benchmark MPMO-MuJoCo and discrete benchmark MP-FTN. Figure 4 illustrates the MPHV and MPSP curves during training for all methods in the MP-HalfCheetah environment. Table 1 reports the evaluation results across all continuous environments. Table 2 presents the comparative performance of MPPN-MORL and other methods in discrete environments. The MPPN-MORL algorithm employs an initial negotiation vector of $(0.5, 0.5)$ . The results for other initial negotiation vectors can be found in Appendix D.2 .

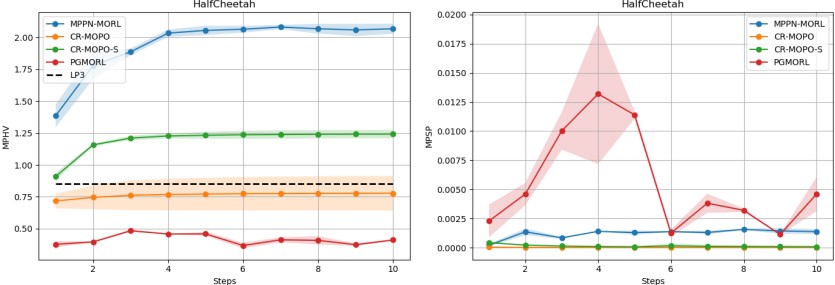

Figure 4: The MPHV and MPSP curve for the MP-HalfCheetah environment. The shaded region represents the standard deviation across six independent experimental runs.

Table 1: Experimental results of MP-MuJoCo environments. Each algorithm was independently executed six times under identical experimental conditions, reporting the mean $\pm$ standard deviation. LP3 and MOAC output a single policy and thus cannot calculate MPSP.

| Environments | Metrics | PGMORL | MOAC | MOCHA | LP3 | CR-MOPO | CR-MOPO-S | MPPN-MORL |
|---|---|---|---|---|---|---|---|---|
| MP-HalfCheetah-v4 | MPHV | 0.411±0.006 | 0.918 | 1.052 | 0.852 | 0.778±0.134 | 1.241±0.032 | **2.067±0.040** |
| | MPSP($10^{-2}$) | 0.458±0.146 | N/A | 0.979 | N/A | **0.001±0.001** | 0.007±0.008 | 0.137±0.021 |
| MP-Walker-v4 | MPHV | 0.000±0.000 | 0.949 | 1.009 | 0.000 | 1.518±0.013 | 0.294±0.037 | **2.897±0.784** |
| | MPSP($10^{-2}$) | 3.136±1.343 | N/A | 53.113 | N/A | 0.195±0.019 | 0.246±0.070 | **0.190±0.088** |
| MP-Hopper-v4 | MPHV | 0.273±0.273 | 0.357 | 1.225 | 0.000 | 1.235±0.034 | 1.376±0.094 | **1.451±0.008** |
| | MPSP($10^{-2}$) | 0.828±0.488 | N/A | 13.999 | N/A | **0.012±0.018** | 0.191±0.032 | 3.003±0.673 |
| MP-Pusher-v4 | MPHV | 0.142±0.029 | 0.401 | 0.589 | 0.063 | 0.398±0.041 | 0.753±0.062 | **0.816±0.007** |
| | MPSP($10^{-2}$) | 6.093±2.731 | N/A | 6.768 | N/A | 0.015±0.002 | **0.014±0.003** | 0.087±0.020 |
| MP-Swimmer-v4 | MPHV | 0.011±0.000 | 0.932 | 0.998 | 0.839 | 0.944±0.035 | 0.995±0.008 | **1.284±0.031** |
| | MPSP($10^{-2}$) | 0.074±0.009 | N/A | 16.354 | N/A | **0.032±0.011** | 0.099±0.005 | 0.753±0.142 |
| MP-Humanoid-v4 | MPHV | 1.720±0.330 | 1.988 | 2.308 | 0.000 | 1.508±0.275 | 1.847±0.194 | **2.761±0.361** |
| | MPSP($10^{-2}$) | 0.381±0.030 | N/A | 5.138 | N/A | 0.002±0.001 | **0.000±0.000** | 0.694±0.032 |

On the MPHV metric, MPPN-MORL achieves the best performance across all MP-MuJoCo environments. This result validates the effectiveness of the proposed algorithm in balancing the interests of multiple parties. For the the discrete benchmark NP-FTN, MPPNMORL achieves the best MPHV at depths 5 and 6, but at depth 7, it performs slightly worse than the PD-MORL method. However, on the MPSP metric, the CR-MOPO algorithm achieves the best performance in three MP-MuJoCo

Table 2: Comparison on the discrete benchmark NP-FTN in terms of MPHV and MPSP.

|  | Fruit Tree Navigation (d=5) | | Fruit Tree Navigation (d=6) | | Fruit Tree Navigation (d=7) | |
|  | MPHV | MPSP | MPHV | MPSP | MPHV | MPSP |
| --- | --- | --- | --- | --- | --- | --- |
| Envelope | 219.150 | **0.020** | 188.770 | **0.020** | 196.840 | **0.020** |
| PD-MORL | 219.150 | **0.020** | 213.200 | **0.020** | **250.960** | **0.020** |
| MPPNMORL | **240.990** | 0.089 | **241.422** | 0.080 | 247.534 | 0.031 |

environments, and the CR-MOPO-S algorithm achieves the best performance in two MP-MuJoCo environments, which can be attributed to their gradient-based optimization that yields a large number of dense policies. It is worth noting that MPPN-MORL exhibits relatively weaker performance on the MPSP metric in both environments. This phenomenon is mainly attributed to the negotiation mechanism of the algorithm, which places greater emphasis on global convergence during optimization, resulting in a sparser distribution of solutions along the Pareto front and consequently higher MPSP values. This indicates that MPPN-MORL has certain limitations in terms of solution distribution.

To verify whether the proposed framework can leverage the advantages of policy gradient, we conducted experiments integrating MOPPO into the framework in Appendix D.5 .

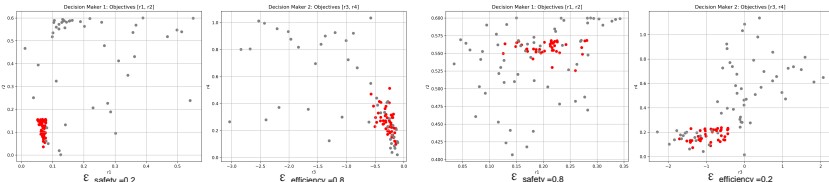

Figure 5: The Pareto policy sets ultimately obtained in MP-HalfCheetah environment with different initial negotiation vectors. The red point means the multi-party Pareto policy.

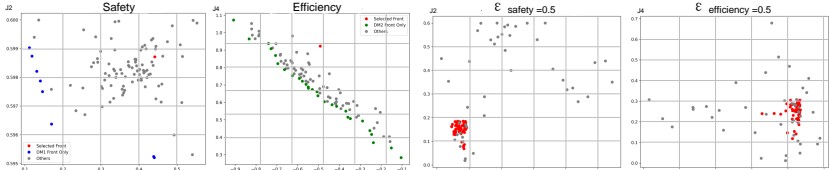

Figure 6: MPNDS algorithm without the MPPN mechanism compares with the MPPN-MORL in MP-HalfCheetah environment.

Across all environments, PGMORL and LP3 exhibit inferior performance, which may be attributed to the difficulty of the predictive model in accurately guiding the policy when the number of objectives is large. MOAC and MOCHA perform policy optimization by dynamically adjusting objective weights, failing to capture the negotiation relationships among decision-making parties. CR-MOPO-S consistently outperforms CR-MOPO, indicating that enforcing safety as a hard constraint limits policy exploration.

Figure 5 compares the Pareto policy sets ultimately obtained with different initial negotiation vectors. It can be observed that by relaxing the constraints of the safety party, significant improvements can be achieved in performance objectives.

By removing the MPPN component, we obtain the multi-objective evolutionary reinforcement learning algorithm MPNDSRL. Figure 6 depicts the Pareto fronts obtained by MPPN-MORL and MP-NDSRL for the two parties in the MP-Halfcheetah environment. MPPN-MORL achieves a better balance between safety and efficiency, and finds a sufficient number of policies while achieving better performance. In contrast, MPNDSRL only finds a very small number of common solutions, which proves that our method can still achieve excellent results when common solutions are scarce.

## 6 RELATED WORK

### 6.1 MORL AND CMORL

MORL tackles tasks with multiple conflicting objectives and mainly includes single-policy and multi-policy approaches. Single-policy methods scalarize multiple rewards into a single objective and apply standard RL to maximize it (Roijers et al., 2013), but they rely on expert-defined preference weights (Van Moffaert et al., 2013; Abdolmaleki et al., 2020) that may vary with real-world conditions. Multi-policy methods approximate the Pareto front by learning a set of policies under different preferences (Roijers et al., 2014; Mossalam et al., 2016; Zuluaga et al., 2016). Typical methods include PGMORL (Xu et al., 2020), which improves efficiency through predictive models and PPO updates but risks local minima; PD-MORL (Basaklar et al., 2023) obtains a unified network covering the entire preference space through single-round training; PA2D-MORL (Hu & Luo, 2024), which uses Pareto ascent directions for automatic optimization and better coverage; and PSL-MORL (Liu et al., 2025), which employs hypernetworks to generate preference-conditioned policies compatible with single-objective RL.

Despite these advances, existing MORL methods optimize multiple objectives only for a single agent and cannot model multi-party interactions or conflicts. Consequently, they fail to capture the negotiation dynamics and collective trade-offs essential in multi-stakeholder scenarios, leading to suboptimal solutions.

CMORL further incorporates safety requirements into multi-objective optimization. LP3 (Huang et al., 2022) jointly learns preferences and policies by treating task rewards and constraint costs as independent objectives. PDOA (Lin et al., 2024) supports offline adaptation under unknown preferences and safety thresholds by learning diverse policies and conservatively estimating preference weights to mitigate violation risks. CR-MOPO (Gu et al., 2025) integrates conflict-aware gradients and hard constraint corrections to ensure safety while efficiently approximating the Pareto front.

Nevertheless, CMORL still focuses on single-agent optimization, lacking mechanisms to model interactions and negotiations among multiple parties. In contrast, MPMORL explicitly captures multi-agent interactions and negotiation dynamics, offering superior modeling capability and adaptability in complex multi-stakeholder environments.

### 6.2 MPMOP

MPMOPs aim to identify mutually optimal solutions for multiple DMs with diverse and often conflicting objectives, a critical challenge in many real-world scenarios. To address this, researchers have developed various MPMOEAs by extending existing MOEA frameworks with ranking and selection mechanisms for multi-party settings. OptMPNDS (Liu et al., 2020) ranks solutions by their worst dominance level across all DMs, while OptMPNDS2 (She et al., 2021) refines this by treating dominance levels from each DM as new objectives and applying a second non-dominated sorting for finer evaluation. A theoretical analysis (Sun et al., 2025) further revealed the inefficiency of traditional MOEAs, especially for NP-hard problems.

Despite the success of MPMOEAs in solving MPMOPs, no prior studies have integrated them into RL. The proposed MPPN-MORL addresses this gap by reformulating MORL as an MPMOP, thereby establishing the first link between these two research domains.

## 7 CONCLUSION

This paper reformulates MORL with safety constraints as a MPMORL problem and proposes an evolutionary algorithm, MPPN-MORL, based on a multi-party Pareto negotiation mechanism. It treats efficiency and safety as independent parties, maintaining separate Pareto fronts for each and merging them via NBMPNDS. This reduces complexity from objective proliferation in traditional MORL. Unlike CMORL, which enforces safety as a hard constraint and strictly limits exploration, MPPN-MORL dynamically adjusts the trade-off between safety and efficiency, producing high-quality compromise solutions. Experimental results demonstrate that across six MP-MuJoCo environments, MPPN-MORL consistently achieves the highest MeanHV and SP metrics, significantly outperforming state-of-the-art MORL and CMORL methods, while exhibiting superior balance and diversity in strategies when handling conflicts between safety and efficiency.

## 8 REPRODUCIBILITY STATEMENT

We have taken extensive efforts to ensure the reproducibility of our work. The proposed algorithms and benchmark implementations have been anonymously submitted as supplementary materials and will be publicly released upon publication.The benchmark environment setups are detailed. These resources collectively enable independent verification and reproduction of our reported results.

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

## A  THE USE OF LARGE LANGUAGE MODELS

We used a large language model to assist in polishing the writing and checking for grammatical issues.

## B  MPPN-SAFEMORL

In Algorithm 2 , we describe the whole procedure of our proposed MPPN-SafeMORL algorithm.

---

**Algorithm 2** Multi-Party NSDE with $\varepsilon$-Dominance and Priority Selection

---

**Input**: Number of iterations $T$, population size $N$, mutation factor $F$, crossover rate $CR$, initial tolerance $\varepsilon_{\text{init}}$, decay rate $\Delta\varepsilon$, objective partitions $\{O_1, O_2\}$
**Output**: Final $\varepsilon$-dominant Pareto front $\mathcal{F}$

 1: Initialize empty population $\mathcal{P} \leftarrow \emptyset$
 2: **for** $i = 1$ to $N$ **do**
 3:     Initialize random policy $\pi_i$ with parameters $\theta_i$
 4:     Evaluate $\pi_i$ to obtain reward vector $\mathbf{r}_i$
 5:     Add $(\pi_i, \theta_i, \mathbf{r}_i)$ to $\mathcal{P}$
 6: **end for**
 7: Initialize tolerance vector $\varepsilon = [\varepsilon_{\text{init}}, \varepsilon_{\text{init}}]$
 8: Compute reference solution $\mathbf{r}_{\text{ref}}$
 9: **for** $t = 1$ to $T$ **do**
10:     $\mathcal{Q} \leftarrow \emptyset$
11:     **for** $i = 1$ to $N$ **do**
12:         Generate offspring parameters $\theta_{\text{trial}}$ using **DE** mutation and crossover with $(F, CR)$
13:         Evaluate offspring to obtain $\mathbf{r}_{\text{trial}}$
14:         Add offspring $(\pi_{\text{trial}}, \theta_{\text{trial}}, \mathbf{r}_{\text{trial}})$ to $\mathcal{Q}$
15:     **end for**
16:     Combine populations: $\mathcal{C} \leftarrow \mathcal{P} \cup \mathcal{Q}$
17:     Extract rewards $\mathcal{R} = \{\mathbf{r}_j\}_{j \in \mathcal{C}}$
18:     Identify $\varepsilon$-dominant fronts:
            1.    Find joint solutions $\mathbf{r}_j$ such that $\mathbf{r}_j$ $\varepsilon$-dominates $\mathbf{r}_{\text{ref}}$ for both DMs.
            2.    If none, update each DM's front separately based on its own objective set $O_k$.
19:     If joint $\varepsilon$-dominant solutions are found: update $\varepsilon \leftarrow \varepsilon * \Delta\varepsilon$
20:     Build next population $\mathcal{P}_{t+1}$:
            1.    Add all individuals from $\varepsilon$-dominant fronts to $\mathcal{P}_{t+1}$
            2.    If $|\mathcal{P}_{t+1}| < N$, fill the remaining slots by applying crowding distance selection on the rest of $\mathcal{C}$
21:     $\mathcal{P} \leftarrow \mathcal{P}_{t+1}$
22: **end for**
23: Extract final front $\mathcal{F}$ from $\mathcal{P}$ based on $\varepsilon$-dominance
24: **return** $\mathcal{F}$

---

# C    EXPERIMENT SETUP DETAILS

For each episode, the reward (or cost) for each objective is computed as the average of the corresponding per-step values over all time steps within that episode.

**MP-Halfcheetah**: In the MP-HalfCheetah environment, the safety party seeks to minimize energy consumption and maintain the stability of the robot's height, whereas the efficiency party aims to maximize forward velocity while mitigating excessive oscillations. Energy consumption is quantified as the squared norm of the action vector:

$$C_e^i = -\alpha_a \|a_{\text{cheetah}}^i\|^2, \tag{14}$$

where $C_e^i$ denotes the energy consumption at time step $i$, $\alpha_a$ is a scaling coefficient, and $a_{\text{cheetah}}^i$ represents the action vector applied to the HalfCheetah at time step $i$.

Height stability is evaluated by the deviation of the robot's height from a target value:

$$C_h^i = \left| H_{\text{cheetah}}^i - H_{\text{target}}^i \right|, \tag{15}$$

where $C_h^i$ is the height stability cost at time step $i$, $H_{\text{cheetah}}^i$ is the actual torso height of the HalfCheetah at time step $i$, and $H_{\text{target}}^i$ is the predefined target height.

Forward velocity is represented by the absolute value of the robot's horizontal velocity:

$$R_x^i = \left| V_x^i \right|, \tag{16}$$

where $R_x^i$ denotes the forward velocity reward at time step $i$, and $V_x^i$ is the horizontal velocity of the HalfCheetah at time step $i$.

Excessive oscillation is penalized using the absolute value of the robot's vertical velocity:

$$R_y^i = -\left|V_y^i\right|, \tag{17}$$

where $R_y^i$ is the oscillation penalty at time step $i$, and $V_y^i$ represents the vertical velocity of the HalfCheetah at time step $i$.

**MP-Hopper**: In the MP-Hopper environment, the safety party aims to minimize the robot's angular deviation around the z-axis and reduce energy consumption, while the efficiency party seeks to maximize forward velocity in the x-direction while minimizing vertical oscillations in the y-direction.

The angular deviation around the z-axis is quantified by the absolute value of the robot's z-axis angle:

$$C_z^i = \left|\Theta_z^i\right|, \tag{18}$$

where $C_z^i$ denotes the angular deviation cost at time step $i$, and $\Theta_z^i$ is the robot's orientation angle around the z-axis at time step $i$.

Energy consumption is measured as the squared norm of the action vector:

$$C_e^i = -\alpha_a |a_{\text{hopper}}^i|^2, \tag{19}$$

where $C_e^i$ denotes the energy consumption at time step $i$, $\alpha_a$ is a scaling coefficient, and $a_{\text{hopper}}^i$ represents the action vector applied to the Hopper at time step $i$.

Forward velocity is represented by the absolute value of the robot's horizontal velocity in the x-direction:

$$R_x^i = \left|V_x^i\right|, \tag{20}$$

where $R_x^i$ denotes the forward velocity reward at time step $i$, and $V_x^i$ is the horizontal velocity of the Hopper at time step $i$.

Vertical oscillation is penalized using the absolute value of the robot's velocity in the y-direction:

$$R_y^i = -\left|V_y^i\right|, \tag{21}$$

where $R_y^i$ is the oscillation penalty at time step $i$, and $V_y^i$ represents the vertical velocity of the Hopper at time step $i$.

**MP-Walker**: In the MP-Walker environment, the safety party aims to minimize the absolute height of the robot's head and reduce the degree of body posture deviation, while the efficiency party seeks to maximize forward velocity in the x-direction while minimizing energy consumption.

The head height cost is quantified by the absolute value of the robot's head height:

$$C_z^i = \left|Z_{\text{head}}^i\right|, \tag{22}$$

where $C_z^i$ denotes the head height cost at time step $i$, and $Z_{\text{head}}^i$ is the vertical height of the Walker's head at time step $i$.

The body posture cost is measured by the absolute deviation of the robot's posture:

$$C_p^i = \left|P_{\text{walker}}^i\right|, \tag{23}$$

where $C_p^i$ denotes the posture deviation cost at time step $i$, and $P_{\text{walker}}^i$ represents the robot's body inclination angle at time step $i$.

Forward velocity is represented by the absolute value of the robot's horizontal velocity in the x-direction:

$$R_x^i = \left|V_x^i\right|, \tag{24}$$

where $R_x^i$ denotes the forward velocity reward at time step $i$, and $V_x^i$ is the horizontal velocity of the Walker at time step $i$.

Energy consumption is quantified as the squared norm of the action vector:

$$C_e^i = -\alpha_a |a_{\text{walker}}^i|^2, \tag{25}$$

where $C_e^i$ denotes the energy consumption at time step $i$, $\alpha_a$ is a scaling coefficient, and $a_{\text{walker}}^i$ represents the action vector applied to the Walker at time step $i$.

**MP-Swimmer**: In the MP-Swimmer environment, the safety party aims to minimize energy consumption and reduce the degree of body oscillation, while the efficiency party seeks to maximize forward velocity in the x-direction while minimizing vertical velocity in the y-direction.

Energy consumption is quantified as the squared norm of the action vector:

$$C_e^i = -\alpha_a |a_{\text{swimmer}}^i|^2, \tag{26}$$

where $C_e^i$ denotes the energy consumption at time step $i$, $\alpha_a$ is a scaling coefficient, and $a_{\text{swimmer}}^i$ represents the action vector applied to the Swimmer at time step $i$.

Body oscillation is measured by the absolute value of the robot's angular velocity:

$$C_o^i = \left| \Omega_{\text{swimmer}}^i \right|, \tag{27}$$

where $C_o^i$ denotes the body oscillation cost at time step $i$, and $\Omega_{\text{swimmer}}^i$ is the angular velocity of the Swimmer at time step $i$.

Forward velocity is represented by the absolute value of the robot's horizontal velocity in the x-direction:

$$R_x^i = \left| V_x^i \right|, \tag{28}$$

where $R_x^i$ denotes the forward velocity reward at time step $i$, and $V_x^i$ is the horizontal velocity of the Swimmer at time step $i$.

Vertical velocity is penalized using the absolute value of the robot's velocity in the y-direction:

$$R_y^i = - \left| V_y^i \right|, \tag{29}$$

where $R_y^i$ is the vertical velocity penalty at time step $i$, and $V_y^i$ represents the vertical velocity of the Swimmer at time step $i$.

**MP-Pusher**: In the MP-Pusher environment, the safety party aims to minimize energy consumption and reduce the velocity of the robot's end-effector, while the efficiency party seeks to minimize the distance between the actuator and the object as well as the distance between the object and the target position.

Energy consumption is quantified as the squared norm of the action vector:

$$C_e^i = -\alpha_a |a_{\text{pusher}}^i|^2, \tag{30}$$

where $C_e^i$ denotes the energy consumption at time step $i$, $\alpha_a$ is a scaling coefficient, and $a_{\text{pusher}}^i$ represents the action vector applied to the Pusher at time step $i$.

The end-effector velocity cost is measured by the absolute value of the end-effector's velocity:

$$C_v^i = \left| V_{\text{end}}^i \right|, \tag{31}$$

where $C_v^i$ denotes the end-effector velocity cost at time step $i$, and $V_{\text{end}}^i$ is the velocity of the Pusher's end-effector at time step $i$.

The actuator-to-object distance is evaluated as the Euclidean distance between the actuator and the object:

$$R_{ao}^i = - \left| P_{\text{actuator}}^i - P_{\text{object}}^i \right|, \tag{32}$$

where $R_{ao}^i$ denotes the actuator-to-object distance reward at time step $i$, $P_{\text{actuator}}^i$ is the position of the actuator at time step $i$, and $P_{\text{object}}^i$ is the position of the object at time step $i$.

The object-to-target distance is evaluated as the Euclidean distance between the object and the target position:

$$R_{ot}^i = - \left| P_{\text{object}}^i - P_{\text{target}}^i \right|, \tag{33}$$

where $R_{ot}^i$ denotes the object-to-target distance reward at time step $i$, $P_{\text{target}}^i$ is the predefined target position, and $P_{\text{object}}^i$ is the object's position at time step $i$.

**MP-Humanoid**: In the MP-Humanoid environment, the safety party aims to minimize energy consumption and reduce contact impact, while the efficiency party seeks to maximize forward velocity in the x-direction and enhance the humanoid's health reward.

Energy consumption is quantified as the squared norm of the action vector:

$$C_e^i = -\alpha_a |a_{\text{humanoid}}^i|^2, \tag{34}$$

where $C_e^i$ denotes the energy consumption at time step $i$, $\alpha_a$ is a scaling coefficient, and $a_{\text{humanoid}}^i$ represents the action vector applied to the Humanoid at time step $i$.

Contact impact is measured by the magnitude of the external contact forces exerted on the humanoid:

$$C_c^i = \left| F_{\text{contact}}^i \right|, \tag{35}$$

where $C_c^i$ denotes the contact impact cost at time step $i$, and $F_{\text{contact}}^i$ represents the contact force vector applied to the Humanoid at time step $i$.

Forward velocity is represented by the absolute value of the humanoid's horizontal velocity in the x-direction:

$$R_x^i = \left| V_x^i \right|, \tag{36}$$

where $R_x^i$ denotes the forward velocity reward at time step $i$, and $V_x^i$ is the horizontal velocity of the Humanoid at time step $i$.

The health reward is quantified by the humanoid's uprightness and stability:

$$R_h^i = H_{\text{humanoid}}^i, \tag{37}$$

where $R_h^i$ denotes the health reward at time step $i$, and $H_{\text{humanoid}}^i$ is the environment-defined health indicator of the Humanoid at time step $i$.

# D  ADDITIONAL EXPERIMENT RESULTS

## D.1  RESULTS OF TOY EXPERIMENT OF CARGOROBOT

The three representative policies illustrate the distinct characteristics of MORL, CMORL, and MP-MORL in the MP-CargoRobot environment. Table 3 shows the representative policies obtained by different algorithms in the toy experiment on the MP-CargoRobot environment.The MORL solution emphasizes overall efficiency, achieving relatively balanced performance across the four objectives, particularly showing strong stability after scaling. In contrast, the CMORL solution reflects the effect of enforcing safety-related constraints: it yields substantially higher energy efficiency, but at the cost of reduced speed and capacity, as expected when prioritizing constraint satisfaction. The MPMORL solution lies between these two extremes. By incorporating multi-party preferences from both the efficiency-oriented DM and the safety-oriented DM, the resulting policy preserves part of the safety advantage while preventing excessive degradation in efficiency, demonstrating a nego-tiated trade-off that neither single-party optimization can obtain. This comparison highlights how multi-party negotiation can lead to solutions capturing balanced compromise among conflicting objectives.

Table 3: Performance comparison across the four objectives in the MP-CargoRobot environment.

| Method | speed | capacity | energy | stability |
|--------|-------|----------|--------|-----------|
| MORL | -0.60 | -0.75 | 0.15 | 1.43 |
| CMORL | -1.50 | -0.98 | 0.75 | 0.60 |
| MPMORL | -1.05 | -1.35 | 0.30 | 1.13 |

## D.2  RESULTS OF DIFFERENT INITIAL NEGOTIATION VECTORS

We conducted experiments with different initial negotiation vectors on the MP-HalfCheetah environment, and the experimental results are presented in the figure 7 .

## D.3  TRAINING TREND OF MPHV AND MPSP

We plotted the trends of MPHV and MPSP in the MP-HalfCheetah environment, presented in the figure 8.

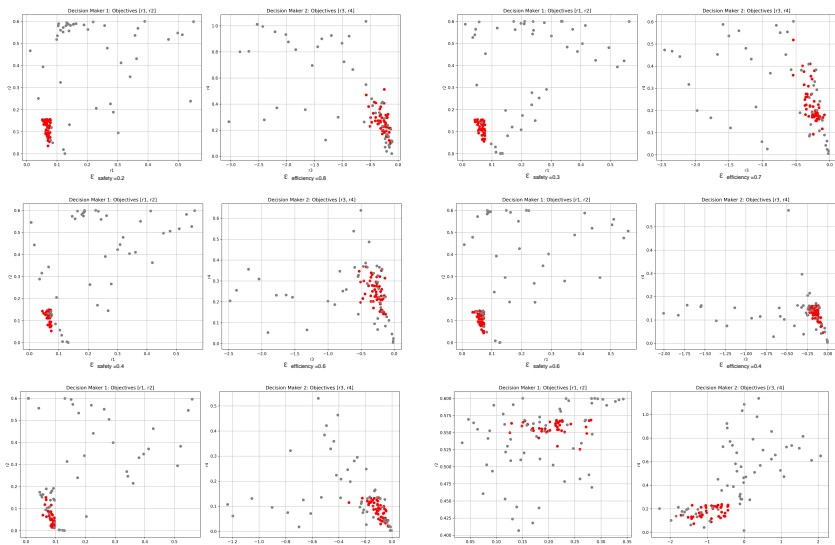

Figure 7: MPPN-MORL with different initial negotiation vectors.

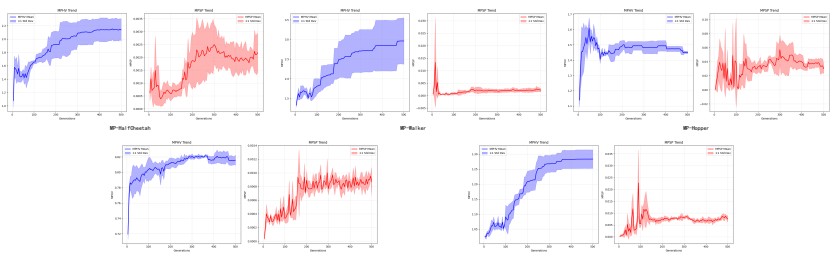

Figure 8: The trends of MPHV and MPSP in the MP-Mujoco environment.

### D.4 RESULTS OF THE MODIFIED JAIN INDEX.

To avoid the misleading effect that the classical Jain's Fairness Index may award a high score when both decision makers receive very small HV values, we adopt a scale-adjusted Jain fairness measure.

For two hypervolume values $HV_A$ and $HV_B$, the classical Jain index is

$$J_{\text{classic}} = \frac{(HV_A + HV_B)^2}{2(HV_A^2 + HV_B^2)}. \tag{38}$$

While this metric captures balance, it fails to penalize cases where $HV_A$ and $HV_B$ are both extremely small. In such cases, $J_{\text{classic}} \approx 1$ even though neither decision maker obtains a meaningful solution set.

To correct this, we introduce a multiplicative scaling factor that depends on the total magnitude of the hypervolumes:

$$S = \frac{HV_A + HV_B}{HV_A + HV_B + \alpha}, \tag{39}$$

where $\alpha > 0$ is a tunable threshold controlling when the overall HV is considered sufficiently large. The adjusted fairness becomes

$$J_{\text{adj}} = J_{\text{classic}} \times \frac{HV_A + HV_B}{HV_A + HV_B + \alpha}. \tag{40}$$

The scaling factor $S$ suppresses inflated fairness values when $HV_A + HV_B$ is very small: If $HV_A + HV_B \to 0$, then $S \to 0$, thus $J_{\text{adj}} \to 0$, preventing false fairness. If $HV_A + HV_B$ is large, then $S \to 1$, and $J_{\text{adj}}$ recovers the classical Jain index.

The hyperparameter $\alpha$ determines how quickly the metric transitions from penalizing small-HV cases to rewarding genuinely balanced performance. It should be chosen according to the typical HV scale of the environment.

This adjusted measure provides a more meaningful fairness evaluation in multi-party multi-objective reinforcement learning by ensuring that fairness is recognized only when both balance *and* solution-set quality are high.

Table 4 presents the adjusted Jain's Fairness Index of different algorithms on the multi-player Mu-JoCo environments.

Table 4: Results of Jain's Fairness Index in MP-MuJoCo environments.

| Environments | PGMORL | MOAC | MOCHA | LP3 | CR-MOPO | CR-MOPO-S | MPPN-MORL |
|---|---|---|---|---|---|---|---|
| MP-HalfCheetah-v4 | 0.344±0.015 | 0.661 | 0.630 | 0.581 | 0.459±0.025 | **0.672±0.010** | 0.618±0.014 |
| MP-Walker-v4 | 0.000±0.000 | 0.646 | 0.520 | 0.000 | 0.411±0.007 | 0.323±0.062 | **0.681±0.037** |
| MP-Hopper-v4 | 0.040±0.040 | 0.706 | 0.200 | 0.000 | 0.703±0.012 | 0.724±0.019 | **0.743±0.002** |
| MP-Pusher-v4 | 0.243±0.018 | 0.531 | 0.280 | 0.140 | 0.257±0.049 | 0.595±0.023 | **0.611±0.002** |
| MP-Swimmer-v4 | 0.015±0.002 | 0.661 | 0.640 | 0.544 | 0.613±0.020 | 0.660±0.001 | **0.685±0.002** |
| MP-Humanoid-v4 | 0.558±0.015 | **0.703** | 0.550 | 0.000 | 0.500±0.012 | 0.493±0.018 | 0.668±0.007 |

### D.5 MPPNMORL COMBINED WITH POLICY GRADIENT ALGORITHM

We integrate MOPPO into the multi-party NSDE loop by performing a policy-gradient refinement step for one selected party. After NSDE generates new candidate policies, MOPPO jointly optimizes that party's full objective vector while leaving the other parties' objectives unchanged. The updated policies are then re-evaluated on all objectives and passed to multi-party non-dominated sorting. This hybrid design allows NSDE to preserve global exploration and multi-party Pareto diversity, while MOPPO provides targeted local improvement for the chosen party, accelerating convergence toward high-quality multi-party Pareto sets. Table 5 shows the experimental results of MPPN-MORL combined with the MOPPO algorithm on the MP-MuJoCo environments. Algorithm 3 details the specific algorithmic procedure of integrating MOPPO into MPPN-MORL. The results demonstrate that the policy gradient information provided by MOPPO effectively aids population evolution, guiding the population toward higher-quality solution sets.

Table 5: Performance of MPPN-MORL with MOPPO across MP-MuJoCo environments.

| Metrics | MP-HalfCheetah-v4 | MP-Walker-v4 | MP-Hopper-v4 | MP-Pusher-v4 | MP-Swimmer-v4 |
|---|---|---|---|---|---|
| MPHV | 2.458 | 4.479 | 1.499 | 1.000 | 25.990 |
| MPSP | 0.0003 | 0.0009 | 4.5782 | 0.0006 | 0.6337 |

---

**Algorithm 3** MP-NSDE with Party-Selective MOPPO

---

1: Initialize population $\mathcal{P}_0$ of policies
2: **for** $g = 1, 2, \ldots, G$ **do**
3:    **NSDE:** Generate offspring $\mathcal{Q}_g$ via mutation and crossover
4:    Evaluate $\mathcal{Q}_g$ on all parties' objectives
5:    **MOPPO:** Select a party $k$ and a subset $\mathcal{S}_g \subseteq \mathcal{Q}_g$
6:    **for** each policy $\pi \in \mathcal{S}_g$ **do**
7:       Collect trajectories with $\pi$
8:       Construct the MOPPO objective using all objectives of party $k$
9:       Update policy: $\pi \leftarrow$ MOPPO_Update$(\pi)$
10:    **end for**
11:    Re-evaluate all updated policies on every party's objectives
12:    **MPNDS:** Perform multi-party non-dominated sorting on $\mathcal{P}_{g-1} \cup \mathcal{Q}_g$
13:    Select next population $\mathcal{P}_g$
14: **end for**
15:
16: **return** multi-party Pareto set $\mathcal{P}_G$

---

### D.6 BEHAVIORAL ANALYSIS OF MULTI-PARTY PARETO POLICIES

The supplementary material contains behavioral visualization GIFs of policies from individual single-party Pareto fronts and the joint two-party Pareto front. These visualizations reveal that policies on the safety party's front tend to move slowly and cautiously, while policies on the efficiency party's front exhibit large, rapid movements. The policies on the joint Pareto front achieve an effective balance between these competing behaviors, demonstrating the negotiation outcome between safety constraints and performance objectives.

### D.7 COMPUTATIONAL EXPENSE ANALYSIS

The table 6 presents the total number of floating-point operations (FLOPs) required for one complete run of three algorithms—MPPNMORL, CR-MOPO, and CR-MOPO-S—in the HalfCheetah environment. As shown, MPPNMORL requires significantly fewer FLOPs than the other two algorithms. This efficiency stems from its use of differential evolution to optimize policy parameters, which avoids the computational overhead associated with policy gradient calculations.

Table 6: Computational cost (FLOPs) of different algorithms on MP-HalfCheetah-v4.

| Metric | MOAC | MOCHA | CR-MOPO | CR-MOPO-S | MPPN-MORL |
|---|---|---|---|---|---|
| FLOPs | $3.65 \times 10^9$ | $2.69 \times 10^{10}$ | $7.89 \times 10^{12}$ | $3.10 \times 10^{13}$ | $8.97 \times 10^5$ |

### D.8 RESULTS OF WILCOXON SIGNED-RANK TEST

To evaluate the statistical significance of MPPN-MORL relative to baseline methods, we employ the Wilcoxon signed-rank test to conduct pairwise comparisons between MPPN-MORL and the better-performing algorithms CR-MOPO-S and CR-MOPO. Table 7 presents the results of the Wilcoxon signed-rank test comparing MPPN-MORL with CR-MOPO and CR-MOPO-S. It can be observed that the improvement achieved by MPPN-MORL on the MPHV metric is statistically significant.

Table 7: Wilcoxon signed-rank test results for MPPN-MORL vs. baseline algorithms(threshold p = 0.05).

| Comparison | W-statistic | p-value | Significant? |
|---|---|---|---|
| **MP-HalfCheetah-v4** | | | |
| MPPN-MORL vs. CR-MOPO | 0.0 | 0.031 | Yes |
| MPPN-MORL vs. CR-MOPO-S | 0.0 | 0.031 | Yes |
| **MP-Walker-v4** | | | |
| MPPN-MORL vs. CR-MOPO | 0.0 | 0.031 | Yes |
| MPPN-MORL vs. CR-MOPO-S | 0.0 | 0.031 | Yes |
| **MP-Hopper-v4** | | | |
| MPPN-MORL vs. CR-MOPO | 0.0 | 0.031 | Yes |
| MPPN-MORL vs. CR-MOPO-S | 0.0 | 0.031 | Yes |
| **MP-Pusher-v4** | | | |
| MPPN-MORL vs. CR-MOPO | 0.0 | 0.031 | Yes |
| MPPN-MORL vs. CR-MOPO-S | 1.0 | 0.062 | No |
| **MP-Swimmer-v4** | | | |
| MPPN-MORL vs. CR-MOPO | 0.0 | 0.031 | Yes |
| MPPN-MORL vs. CR-MOPO-S | 0.0 | 0.031 | Yes |
| **MP-Humanoid-v4** | | | |
| MPPN-MORL vs. CR-MOPO | 0.0 | 0.031 | Yes |
| MPPN-MORL vs. CR-MOPO-S | 0.0 | 0.031 | Yes |

# E DETAILED PROOFS OF THEORETICAL PROPERTIES

## E.1 THEORETICAL JUSTIFICATION OF $\varepsilon$-NEGOTIATION CONVERGENCE

We now provide the theoretical justification addressing why the proposed $\varepsilon$-dominance negotiation, combined with a dynamic shrinking mechanism, progressively guides the population toward superior multi-party Pareto solutions. This analysis formally links the shrinking tolerance $\varepsilon$ to the convergence toward the true joint Pareto front.

Our proof framework relies on (i) the monotonic nesting of $\varepsilon$-acceptable solution sets and (ii) a time-scale separation between the evolutionary search (population mixing) and the negotiation process ($\varepsilon$-shrinking).

### E.1.1 NESTED $\varepsilon$-ACCEPTABLE SETS

We first formalize the set of solutions that are acceptable to a single party, and then define the joint set as the consensus (intersection) of these individual sets, directly matching the logic in Algorithm 1.

**Definition E.1** (Party-wise $\varepsilon$-Acceptable Set). Let $\mathcal{X}$ be the solution space and $X_{\text{ref}} \in \mathcal{X}$ be a reference solution. For a party $k \in \{1, 2\}$, given its $m_k$ objectives $\{f_{k,1}, \ldots, f_{k,m_k}\}$ and a scalar negotiation tolerance $\varepsilon_k \geq 0$, the *party-wise $\varepsilon$-acceptable set* $S_k(\varepsilon_k)$ is defined as:

$$S_k(\varepsilon_k) \triangleq \{X \in \mathcal{X} \mid f_{k,i}(X) \leq f_{k,i}(X_{\text{ref}}) + \varepsilon_k, \quad \forall i \in \{1, \ldots, m_k\}\} \tag{41}$$

$S_k(\varepsilon_k)$ contains all solutions that party $k$ finds acceptable, allowing a uniform tolerance $\varepsilon_k$ across all its local objectives relative to the reference.

**Definition E.2** (Joint $\varepsilon$-Acceptable Set). Given the party-wise tolerances $\boldsymbol{\varepsilon} = [\varepsilon_1, \varepsilon_2]^\top$, the *joint $\varepsilon$-acceptable set* $S(\boldsymbol{\varepsilon})$ is the set of solutions mutually acceptable to all parties. This set is the intersection of the individual party-wise sets:

$$S(\boldsymbol{\varepsilon}) \triangleq S_1(\varepsilon_1) \cap S_2(\varepsilon_2) = \bigcap_{k \in \{1,2\}} S_k(\varepsilon_k) \tag{42}$$

This set $S(\boldsymbol{\varepsilon})$ is the formal representation of the $\mathcal{F}_{joint}$ (Joint $\varepsilon$-front) sought by Algorithm 1.

This formulation leads to a crucial property: as the negotiation becomes stricter (i.e., $\boldsymbol{\varepsilon}$ shrinks), the set of mutually acceptable solutions becomes monotonically smaller and nested.

**Lemma 1** (Monotonicity of Nested Sets). Let $\boldsymbol{\varepsilon}_a = [\varepsilon_{1,a}, \varepsilon_{2,a}]^\top$ and $\boldsymbol{\varepsilon}_b = [\varepsilon_{1,b}, \varepsilon_{2,b}]^\top$ be two tolerance vectors. If $\boldsymbol{\varepsilon}_a \geq \boldsymbol{\varepsilon}_b$ (component-wise, i.e., $\varepsilon_{k,a} \geq \varepsilon_{k,b}$ for all $k$), then their corresponding joint acceptable sets are nested:

$$S(\boldsymbol{\varepsilon}_a) \supseteq S(\boldsymbol{\varepsilon}_b) \tag{43}$$

*Proof.* We first show monotonicity for each party $k$. Let $X \in S_k(\varepsilon_{k,b})$. By Definition 41, $f_{k,i}(X) \leq f_{k,i}(X_{\text{ref}}) + \varepsilon_{k,b}$ for all $i \in \{1, \ldots, m_k\}$. Since $\varepsilon_{k,a} \geq \varepsilon_{k,b}$, it follows that $f_{k,i}(X) \leq f_{k,i}(X_{\text{ref}}) + \varepsilon_{k,b} \leq f_{k,i}(X_{\text{ref}}) + \varepsilon_{k,a}$. This implies $X \in S_k(\varepsilon_{k,a})$. Thus, $S_k(\varepsilon_{k,a}) \supseteq S_k(\varepsilon_{k,b})$ for each $k$.

Now, let $X$ be an arbitrary solution in the joint set $S(\boldsymbol{\varepsilon}_b)$. By Definition 42, $X \in S_k(\varepsilon_{k,b})$ for all $k \in \{1, 2\}$. From our first step, we know $S_k(\varepsilon_{k,a}) \supseteq S_k(\varepsilon_{k,b})$. Therefore, $X \in S_k(\varepsilon_{k,a})$ for all $k$. By Definition 42 again, $X$ must be in the intersection of these sets: $X \in \bigcap_k S_k(\varepsilon_{k,a})$, which means $X \in S(\boldsymbol{\varepsilon}_a)$. This proves $S(\boldsymbol{\varepsilon}_a) \supseteq S(\boldsymbol{\varepsilon}_b)$. $\square$

### E.1.2 LAYERED CONVERGENCE VIA TIME-SCALE SEPARATION

The MPPN algorithm dynamically shrinks $\boldsymbol{\varepsilon}$ to $\boldsymbol{\varepsilon}'$ only when a sufficient number of solutions are found in the current joint set $S(\boldsymbol{\varepsilon})$ (i.e., $|\mathcal{F}_{\text{joint}}| \geq \tau$). This mechanism relies on the following standard assumptions regarding the evolutionary dynamics.

- **Assumption 1 (Irreducibility):** For any fixed $\boldsymbol{\varepsilon}$, any solution $X \in S(\boldsymbol{\varepsilon})$ can be generated by the EA operators (e.g., differential evolution) from any population $P_t$ in a finite number of generations with non-zero probability.

- **Assumption 2 (Retention):** The selection mechanism (elitism and non-dominated sorting) ensures that if a solution $X \in S(\varepsilon)$ is found, at least one representative $X' \in S(\varepsilon)$ is retained in the next generation's population with high probability.

- **Assumption 3 (Time-Scale Separation):** For any fixed $\varepsilon$, the EA has a characteristic *mixing time*, $T_{\text{mix}}(\varepsilon)$, within which the population $P_t$ is expected to find and provide representative coverage of the set $S(\varepsilon)$. The negotiation mechanism only shrinks $\varepsilon$ at time $T_{\text{shrink}}$ (when $|\mathcal{F}_{\text{joint}}| \geq \tau$), and we assume $T_{\text{shrink}} > T_{\text{mix}}(\varepsilon)$.

These assumptions allow us to prove that the population is progressively guided into stricter subsets of the solution space.

**Theorem 1** (Layered Convergence to Stricter Pareto Sets). Let the sequence of tolerance vectors generated by the shrinking mechanism be $\varepsilon_0, \varepsilon_1, \ldots, \varepsilon_T$ such that $\varepsilon_0 \geq \varepsilon_1 \geq \cdots \geq \varepsilon_T$. Let $P_j$ be the population that triggers the $j$-th shrink (i.e., $P_j$ contains at least $\tau$ solutions from $S(\varepsilon_j)$). Under Assumptions 1-3, the population $P_t$ converges in probability to the final, strictest acceptable set $S(\varepsilon_T)$:

$$\lim_{t \to \infty} P_t \subseteq \bigcap_{j=0}^{T} S(\varepsilon_j) = S(\varepsilon_T) \tag{44}$$

*Proof.* We proceed by induction on the negotiation steps $j = 0, 1, \ldots, T$.

**Base Case ($j = 0$):** The algorithm begins with a large, lenient tolerance $\varepsilon_0$. By Assumptions 1 and 2, the EA explores the solution space $\mathcal{X}$. By Assumption 3 (Time-Scale Separation), the algorithm runs for sufficient time ($T_{\text{mix}}(\varepsilon_0)$) to find and populate the set $S(\varepsilon_0)$ before the shrink condition is met. At time $t_0 = T_{\text{shrink}}(\varepsilon_0)$, the population $P_{t_0}$ provides representative coverage of $S(\varepsilon_0)$.

**Inductive Step:** Assume at negotiation step $j$, the algorithm has run for time $t_j$ and the population $P_{t_j}$ provides representative coverage of $S(\varepsilon_j)$. At time $t_j$, the condition $|\mathcal{F}_{\text{joint}}| \geq \tau$ is met, and the tolerance is shrunk to $\varepsilon_{j+1}$.

By Lemma 1 (Monotonicity), we know that $S(\varepsilon_{j+1}) \subseteq S(\varepsilon_j)$.

The population $P_{t_j}$ is already concentrated within $S(\varepsilon_j)$. The EA search is now "warm-started" and focused on finding solutions that satisfy the new, stricter criteria of $S(\varepsilon_{j+1})$. Since $S(\varepsilon_{j+1})$ is a non-empty subset of the region $S(\varepsilon_j)$ already discovered, the search is guided toward this higher-quality region.

By Assumption 3, the algorithm again runs for at least $T_{\text{mix}}(\varepsilon_{j+1})$ generations. Assumptions 1 and 2 ensure the EA will find and retain solutions within this new, smaller set $S(\varepsilon_{j+1})$. At time $t_{j+1} = t_j + T_{\text{shrink}}(\varepsilon_{j+1})$, the population $P_{t_{j+1}}$ will provide representative coverage of $S(\varepsilon_{j+1})$.

By induction, the population $P_t$ is proven to follow the sequence of strictly nested sets $S(\varepsilon_0) \supseteq S(\varepsilon_1) \supseteq \cdots \supseteq S(\varepsilon_T)$. The final population $P^*$ is thus contained within the strictest set achieved, $S(\varepsilon_T)$.

**Implication:** This layered convergence demonstrates that the $\varepsilon$-shrinking mechanism is not merely finding an approximation $S(\varepsilon)$ for a fixed $\varepsilon$. Instead, it actively *guides* the evolutionary search by iteratively tightening the acceptance criteria (as $\varepsilon$ shrinks), forcing the population to converge from a broad, lenient set of compromises toward the multi-party Pareto front. □

### E.2 ENSURING HARD SAFETY CONSTRAINTS

In many real-world safe reinforcement learning scenarios, the safety requirements of one party may represent *hard constraints* that must never be violated. Formally, let the safety party be denoted as $k = 2$, and let its constraint function be $c(X)$ with a mandatory threshold $d$. The feasible region is thus

$$\Pi_{\text{safe}} \triangleq \{X \in \mathcal{X} \mid c(X) \leq d\}. \tag{45}$$

In this subsection, we show that the MPPN framework can enforce these non-relaxable constraints simply by setting the tolerance vector to $\varepsilon = (1, 0)$.

**Lemma 2** (Safety Feasibility of $S_2(0)$)**.** For the safety party $k = 2$, if $\varepsilon_2 = 0$, then its acceptable set coincides exactly with the hard-constrained feasible region:

$$S_2(0) = \Pi_{\text{safe}}. \tag{46}$$

*Proof.* For party 2, the acceptable set from Definition 41 becomes

$$S_2(0) = \{X \in \mathcal{X} \mid f_{2,i}(X) \le f_{2,i}(X_{\text{ref}}), \ \forall i\}. \tag{47}$$

Since the safety objective $f_{2,i}(\cdot)$ represents constraint violations (i.e., $f_{2,i}(X) = c(X)$ or monotone transformations of $c(X)$), the condition

$$f_{2,i}(X) \le f_{2,i}(X_{\text{ref}})$$

enforces that no additional violation is allowed beyond the reference, and under the assumption that $X_{\text{ref}}$ itself satisfies $c(X_{\text{ref}}) \le d$, this is equivalent to

$$c(X) \le d, \tag{48}$$

which is exactly the feasible region $\Pi_{\text{safe}}$. Therefore, $S_2(0) = \Pi_{\text{safe}}$. $\square$

We now show that when $\varepsilon = (1, 0)$, the joint acceptable set collapses to the safety-feasible region, regardless of the efficiency party's tolerance.

**Theorem 2** (Joint Acceptable Set under Hard Safety Constraint)**.** Let $\varepsilon = (\varepsilon_1, \varepsilon_2) = (1, 0)$. Then the joint acceptable set satisfies

$$S(\varepsilon) = S_1(1) \cap S_2(0) = S_1(1) \cap \Pi_{\text{safe}} \subseteq \Pi_{\text{safe}}. \tag{49}$$

That is, every jointly acceptable solution is guaranteed to obey the hard safety constraint.

*Proof.* From Definition 42, the joint set is the intersection of party-wise constraints:

$$S(\varepsilon) = S_1(1) \cap S_2(0).$$

By Lemma 2, $S_2(0) = \Pi_{\text{safe}}$. Thus,

$$S(\varepsilon) = S_1(1) \cap \Pi_{\text{safe}}.$$

Since $S_1(1) \subseteq \mathcal{X}$, their intersection is always a subset of $\Pi_{\text{safe}}$. Hence,

$$S(\varepsilon) \subseteq \Pi_{\text{safe}},$$

which completes the proof. $\square$

The following theorem establishes that the layered convergence result from the previous subsection remains valid under the hard-constraint case.

**Theorem 3** (Layered Convergence within Hard-Constrained Feasible Region)**.** Consider the shrinking sequence $\varepsilon_0 \ge \varepsilon_1 \ge \cdots \ge \varepsilon_T$ with $\varepsilon_j = (1, 0)$ for all $j$. Under Assumptions 1–3, the population converges in probability to the strictest joint acceptable set contained within the hard-constrained feasible region:

$$\lim_{t \to \infty} P_t \subseteq \bigcap_{j=0}^{T} S(\varepsilon_j) = S(\varepsilon_T) \subseteq \Pi_{\text{safe}}. \tag{50}$$

*Proof.* Since $\varepsilon_2 = 0$ for all $j$, each joint acceptable set satisfies

$$S(\varepsilon_j) = S_1(\varepsilon_{1,j}) \cap \Pi_{\text{safe}} \subseteq \Pi_{\text{safe}}.$$

By Lemma 1 (monotonicity), the sequence is nested:

$$S(\varepsilon_0) \supseteq S(\varepsilon_1) \supseteq \cdots \supseteq S(\varepsilon_T).$$

The proof of Theorem 2 (Layered Convergence) applies verbatim within this restricted domain by simply restricting the search space from $\mathcal{X}$ to $\Pi_{\text{safe}}$. Assumptions 1–3 continue to hold in the restricted domain because irreducibility, retention, and time-scale separation are properties of the evolutionary operators and selection rules, not of the particular tolerance values.

Therefore, the population converges to $S(\varepsilon_T)$, which is itself a subset of $\Pi_{\text{safe}}$. $\square$

These results show that when one party represents a non-negotiable safety requirement, simply setting its tolerance to $\varepsilon_2 = 0$ forces the entire MPPN negotiation and evolutionary search to remain strictly within the hard-constrained feasible region, while still benefiting from the layered guidance induced by iterative $\varepsilon$-shrinking on the efficiency side. Thus, the MPPN framework naturally accommodates hard safety constraints without altering its algorithmic structure.

### E.3 EPSILON-SHRINKING LEADS TO IMPROVED MULTI-PARTY PARETO SOLUTIONS

Beyond the layered convergence shown in Theorem 2, we now provide a formal justification that the $\varepsilon$-shrinking negotiation mechanism in MPPN yields *better multi-party Pareto solutions* as the negotiation progresses. Our analysis is grounded in two complementary properties:

1. the monotonic nesting of joint acceptable sets $S(\varepsilon_0) \supseteq S(\varepsilon_1) \supseteq \cdots$, and
2. the decrease in solution-space complexity (as measured by metric covering numbers), which improves the probability that an evolutionary algorithm discovers high-quality solutions under fixed computational resources.

This result shows that the shrinking of $\varepsilon$ does not merely tighten" acceptance criteria, but actively improves the quality of the resulting negotiated Pareto sets.

#### E.3.1 SOLUTION-SPACE COMPLEXITY OF JOINT ACCEPTABLE SETS

Let $\mathcal{Y}$ denote the objective-space image of the solution space $\mathcal{X}$ under the joint objective vector. For any $\delta > 0$, define the standard metric covering number $N_{\mathrm{cov}}(A, \delta)$ as the minimum number of closed balls of radius $\delta$ required to cover a set $A \subseteq \mathcal{Y}$.

We operate under one mild regularity condition:

**Definition E.3** (Non-Degeneracy). A joint acceptable set $S(\varepsilon)$ is said to be non-degenerate if it is not contained entirely within a lower-dimensional manifold of $\mathcal{Y}$; equivalently, its covering number satisfies $N_{\mathrm{cov}}(S(\varepsilon), \delta) < \infty$ for all $\delta > 0$.

Under this condition, shrinking $\varepsilon$ reduces the size and complexity of $S(\varepsilon)$:

**Lemma 3** (Strict Decrease in Covering Number Under Shrinking). Let $\varepsilon_a \geq \varepsilon_b$ component-wise, and assume $S(\varepsilon_b)$ is non-degenerate. If at least one inequality is strict, then for all $\delta > 0$:

$$N_{\mathrm{cov}}(S(\varepsilon_b), \delta) \; > \; N_{\mathrm{cov}}(S(\varepsilon_a), \delta). \tag{51}$$

*Proof.* From Lemma 1 (nested sets), $S(\varepsilon_b) \subsetneq S(\varepsilon_a)$. Since $S(\varepsilon_b)$ is non-degenerate, removing a region of strictly positive local measure necessarily increases the minimum number of $\delta$-balls needed to cover the remainder (**?**). Thus $N_{\mathrm{cov}}$ is strictly larger for $S(\varepsilon_b)$. $\square$

The covering number $N_{\mathrm{cov}}$ is therefore an intrinsic measure of the difficulty" of discovering good solutions in $S(\varepsilon)$. The next subsection connects this complexity to the performance of the evolutionary search.

#### E.3.2 DISCOVERY PROBABILITY IMPROVES AS $\varepsilon$ SHRINKS

Let $P_t$ denote the population of the EA at generation $t$, and let $\mathcal{F}_{\mathrm{joint}}(t)$ denote its approximation of $S(\varepsilon)$. For a fixed computational budget (population size $N$ and generation budget $T$), the EA's ability to discover and cover $S(\varepsilon)$ depends on its ability to sample at least one solution in each $\delta$-ball of the covering.

The following lemma formalizes this connection.

**Lemma 4** (Discovery Probability and Covering Number). Under Assumptions 1–3, there exists a constant $c > 0$ such that for any $\delta > 0$ and any joint acceptable set $S(\varepsilon)$, the probability that $P_T$ contains at least one point in every $\delta$-ball of a minimal cover of $S(\varepsilon)$ satisfies:

$$\mathbb{P}[d_H(\mathcal{F}_{\mathrm{joint}}(T), S(\varepsilon)) \leq \delta] \; \geq \; 1 - N_{\mathrm{cov}}(S(\varepsilon), \delta) \exp(-cNT). \tag{52}$$

*Proof.* Since each $\delta$-ball receives a sample with probability at least $p_\delta \geq 1 - \exp(-cN)$ due to Assumptions 1 (irreducibility) and 2 (retention), the probability that at least one ball in the cover remains uncovered is bounded via a union bound, giving the stated expression. $\square$

Combining Lemma 3 and Lemma 4 yields the main theoretical result.

**Theorem 4** (Epsilon-Shrinking Improves Multi-Party Pareto Quality). Consider two tolerance vectors $\varepsilon_a \geq \varepsilon_b$ with at least one strict inequality. Let the EA run under the same computational budget $(N, T)$ for both tolerance choices. Then for any $\delta > 0$:

$$\mathbb{P}\Big[d_H\Big(\mathcal{F}_{\text{joint}}^{(b)}(T), S(\varepsilon_b)\Big) \leq \delta\Big] \tag{53}$$

$$> \mathbb{P}\Big[d_H\Big(\mathcal{F}_{\text{joint}}^{(a)}(T), S(\varepsilon_a)\Big) \leq \delta\Big], \tag{54}$$

where $\mathcal{F}_{\text{joint}}^{(j)}$ denotes the joint front under tolerance $\varepsilon_j$.

Consequently, the expected quality of the obtained multi-party Pareto set (strictly) improves:

$$\mathbb{E}\Big[\text{MPHV}\big(\mathcal{F}_{\text{joint}}^{(b)}(T)\big)\Big] > \mathbb{E}\Big[\text{MPHV}\big(\mathcal{F}_{\text{joint}}^{(a)}(T)\big)\Big]. \tag{55}$$

*Proof.* By Lemma 3, shrinking $\varepsilon$ strictly increases the covering number:

$$N_{\text{cov}}\big(S(\varepsilon_b), \delta\big) > N_{\text{cov}}\big(S(\varepsilon_a), \delta\big).$$

Substituting these into the probability bound of Lemma 4, we observe that the larger covering number *strictly increases* the failure term:

$$N_{\text{cov}}\big(S(\varepsilon), \delta\big) \exp(-cNT).$$

Thus the success probability of covering the stricter set $S(\varepsilon_b)$ under identical computational resources is strictly higher. Since MPHV is monotone under Hausdorff improvement, its expected value increases accordingly. $\square$

This theorem shows that the $\varepsilon$-shrinking process of MPPN does not merely guide convergence (Theorem 2); it *improves the quality* of the resulting multi-party Pareto set by reducing solution-space complexity and increasing the probability that the EA identifies the most valuable tradeoffs shared by all parties.

