# OpenReview forum: "Safe Multi-Objective Reinforcement Learning via Multi-Party Pareto Negotiation"
_ICLR.cc/2026/Conference — Submitted to ICLR 2026_

### Official Review · Reviewer_5Y92 · 2025-10-28

**Soundness:** 3
**Presentation:** 2
**Contribution:** 3
**Rating:** 4
**Confidence:** 4

**Summary:**

This work addresses safe multi-objective reinforcement learning within a multi-party negotiation framework, rather than treating safety as an additional objective. This approach enables the search for a consensus-based, multi-party Pareto-optimal set without enlarging the objective space. Within this framework, the authors propose a multi-party Pareto negotiation (MPPN) strategy.

**Strengths:**

1. Formulating safe multi-objective reinforcement learning as a multi-party negotiation problem is, to the best of my knowledge, novel, interesting, and practically valuable.

2. The overall method design is generally reasonable and coherent.

**Weaknesses:**

1. Algorithm 2 appears to update policy parameters solely through DE mutation, without using policy gradients or value-based guidance. This raises questions about efficiency and whether the learning signal may be too sparse, so it is also unclear if this approach can still be considered “reinforcement learning.” The observed improvements may primarily result from the multi-party-specific evaluation metrics, whereas baseline methods are not designed for a multi-party setting.

2. It would be helpful to present the learned behaviors and analyze how they relate to multi-party Pareto optimality.

3. The policy indices in Figure 2 do not seem to correspond with the text description in lines 180–185.

**Questions:**

1. Does it update policy parameters solely through DE mutation?
2. How could policy gradients be incorporated to provide denser and more informative updates?
3. Does the method learn meaningful multi-party Pareto behaviors in complex environments, such as humanoid tasks?

---

> ### Author Response · Authors · 2025-11-28
> **Response to Reviewer 5Y92**
>
> We sincerely thank the reviewer for their time and valuable comments, which have greatly helped us improve the quality of our study.
>
> In Appendix D.5, we present an approach that alternates differential evolution with MOPPO according to Algorithm 3.We have added theoretical analysis in Section 4 of the paper, with corresponding proofs provided in Appendix E. In our comparative experiments, we have included the latest state-of-the-art MORL algorithms, MOAC and MOCHA. Additionally, we have extended our evaluation with comparative experiments of MPPN-MORL in the discrete Fruit Tree Navigation environment.
>
> > W1: Algorithm 2 appears to update policy parameters solely through DE mutation, without using policy gradients or value-based guidance. This raises questions about efficiency and whether the learning signal may be too sparse, so it is also unclear if this approach can still be considered “reinforcement learning.” The observed improvements may primarily result from the multi-party-specific evaluation metrics, whereas baseline methods are not designed for a multi-party setting.
>
> A1: Algorithm 2 indeed relies solely on the DE mutation operator to update the policy parameters, as evolutionary algorithms are better aligned with the multi-party framework we propose. In our paper, the problem is formulated as a Markov Decision Process, wherein the policy interacts with the environment to maximize expected cumulative reward. Although DE is an evolutionary algorithm, its application in optimizing neural network policies for sequential decision-making tasks does not compromise the reinforcement learning framework of MPPNMORL.
>
> Standard MORL metrics do not account for conflicts among the negotiating parties. The MPHV metric was specifically designed to penalize solutions that satisfy one party while completely failing to meet the objectives of another.
>
> We have also compared our method against CR-MOPO and LP3, which represent state-of-the-art constrained multi-objective reinforcement learning approaches for handling safety constraints. These baselines optimize the same objectives (safety + efficiency). The fact that MPPN-MORL outperforms them on the MPHV metric demonstrates that our negotiation-based approach achieves a better trade-off and yields a superior distribution along the Pareto front, rather than merely exploiting loopholes in the evaluation metric.
>
> > W2: It would be helpful to present the learned behaviors and analyze how they relate to multi-party Pareto optimality.
>
> A2: We have visualized the behaviors of multi-party Pareto policies in the HalfCheetah environment and compared them with Pareto policies optimized for a single party only; these visualizations are provided as GIFs in the supplementary materials. In Appendix D.6, we provide a detailed analysis of the behavioral differences between the multi-party Pareto policies and those optimized solely for a single party.
>
> > W3: The policy indices in Figure 2 do not seem to correspond with the text description in lines 180–185.
>
> A3: Thank you for your careful reading; we have ensured correct index alignment.
>
> > Q1: Does it update policy parameters solely through DE mutation?
>
> A1: Algorithm 2 indeed relies solely on the differential evolution (DE) mutation operator to update the policy parameters, as evolutionary algorithms are better suited to the multi-party framework we propose.
>
> > Q2: How could policy gradients be incorporated to provide denser and more informative updates?
>
> A2: In the newly added Appendix D.5, we present an approach that alternates differential evolution with MOPPO according to Algorithm 3. Experimental results across multiple environments demonstrate that incorporating MOPPO improves both algorithmic efficiency and policy density in most settings, leading to the discovery of superior policies.
>
> Table R1 shows the performance of MPPN-MORL with MOPPO across MP-MuJoCo environments.
>
> **Table R1**: Performance of MPPN-MORL with MOPPO across MP-MuJoCo environments.
> | Metrics               | MP-HalfCheetah-v4 | MP-Walker-v4 | MP-Hopper-v4 | MP-Pusher-v4 | MP-Swimmer-v4 |
> |-----------------------|-------------------|--------------|--------------|--------------|---------------|
> | MPHV                  | 2.458             | 4.479        | 1.499        | 1.000        | 25.990        |
> | MPSP                  | 0.0003            | 0.0009       | 4.5782       | 0.0006       | 0.6337        |
>
>
> > Q3: Does the method learn meaningful multi-party Pareto behaviors in complex environments, such as humanoid tasks?
>
> A3: The MP-Humanoid environment constitutes a high-dimensional control task. In this complex setting, MPPN-MORL achieves a multi-party hypervolume (MPHV) of 2.761 ± 0.361, significantly outperforming all baseline methods. Visualizations of the learned policies further confirm that our approach successfully identifies solutions that effectively balance the objectives of the safety-oriented and efficiency-oriented parties.

---

> > ### Comment · Reviewer_5Y92 · 2025-11-28
> >
> > Thanks, I do think the concept of this work is interesting and the additional results look promising. I will raise my score to 6

---

> > > ### Author Response · Authors · 2025-11-29
> > > **Response to Reviewer 5Y92**
> > >
> > > Thank you for your feedback and for considering our work.We will continue to address any remaining concerns and refine our contributions to further improve the quality and impact of our research.

---

### Official Review · Reviewer_mHFd · 2025-10-31

**Soundness:** 3
**Presentation:** 3
**Contribution:** 2
**Rating:** 6
**Confidence:** 3

**Summary:**

This paper reconceptualizes Safe MORL as a multi-party negotiation problem, where the safety objectives and efficiency objectives are treated as separate multi-objective decision parties rather than as additional objectives in a single objective space.
Building on this idea, they develop a negotiation-driven evolutionary framework, MPPN-MORL, which integrates multi-party Pareto negotiation into policy search without increasing the dimension of the objective space. The algorithm incorporates an ε-dominance criterion to enable negotiation into evolutionary search. The idea offers a novel and well-motivated perspective on safe MORL.

**Strengths:**

The paper presents a novel conceptual formulation of safe MORL as a multi-party negotiation process, which provides a fresh perspective on balancing safety and efficiency.

The proposed MPPN-MORL framework is well-motivated, integrating negotiation principles with evolutionary policy search in a coherent way.

The use of an ε-dominance based negotiation rule and differential evolution operators is clearly described and logically connected to the goal of efficient compromise among objectives (see however below).

It is possible that the algorithm respects user-specified preferences over both performance and safety, preserves diversity in the solution set, and promotes fairness across parties.

**Weaknesses:**

The paper claims adaptability, diversity, and fairness in the learned policy set, but these aspects are not directly analyzed or supported by quantitative experiments. Including such evidence would strengthen the empirical evaluation.

The scalability and computational cost of negotiation among multiple parties are not extensively discussed, which may limit understanding of its practical applicability.

Fig. 1 may be excellent for use in a talk, but in a collection containing several contributions on MORL, there is no need to start from this level.

Definition 3.1 does not define Pareto Dominance, but Pareto Dominance w.r.t a DM. This should be stated in the beginning in brackets.

In Table 1, the proposed method is referred to as “MPNN”, while the paper elsewhere uses “MPPN.” Please check whether this is a typo.

Please clarify what the $x$- and $y$-coordinates in Figure 5 represent, including their units and scales? It is important as it is not clear from the caption or text how Figure 5 is obtained or what it is intended to show.

A problem is the need to choose two ε thresholds (for performance and safety) which is at odds with the idea of MORL where the  decision whether a criterion is more less strict is left for the user for after the optimization, while here a related decision is to be made before the start of MORL, so that it is questionable whether the MORL framework has to be used here in the first place or whether already a scalarization is sufficient. I understand that there is theoretical difference between the preference coefficients and the ε thresholds, but it will be difficult to explain this to any users.

**Questions:**

The paper notes that MPPN-MORL has certain limitations in terms of solution distribution. Could the authors elaborate on what causes this limitation, and whether it relates to the ε-dominance mechanism or the negotiation dynamics?

How would you treat safety constraints that cannot be expressed as objectives?

Wouldn't in the case of a safety-critical application a hierarchical approach be useful? I.e. why should unsafe regions be explored at all? If this is in some cases justifiable, then such a justification needs to be discussed already here.

Can you define a Multi-party Pareto Front?

**Details Of Ethics Concerns:**

The algorithm promises safety, diversity and fairness, which may not necessarily meet the criteria of any possible application. Although no problem is visible at this stage, a note of caution be may useful especially as the assessment of these properties is improvable (see above) already for the applications that are considered here.

---

> ### Author Response · Authors · 2025-11-28
> **Response to Reviewer mHFd (1/2)**
>
> We sincerely thank the reviewer for their time and valuable comments, which have greatly helped us improve the quality of our study.
>
> We have added theoretical analysis in Section 4 of the paper, with corresponding proofs provided in Appendix E. In our comparative experiments, we have included the latest state-of-the-art MORL algorithms, MOAC and MOCHA. Additionally, we have extended our evaluation with comparative experiments of MPPN-MORL in the discrete Fruit Tree Navigation environment. In Appendix D.4, we provide an adapted version of Jain’s Fairness Index that jointly evaluates both fairness and policy quality.
>
> > W1: The paper claims adaptability, diversity, and fairness in the learned policy set, but these aspects are not directly analyzed or supported by quantitative experiments. Including such evidence would strengthen the empirical evaluation.
>
> A1: In Appendix D.4, we provide an adapted version of Jain’s Fairness Index that jointly evaluates both fairness and policy quality. The adaptability of MPPN-MORL to the preferences among multiple parties can be assessed through multiple runs of the algorithm. Table R1 shows the adapted results of Jain’s Fairness Index in MP-MuJoCo environments.
>
> **Table R1**: Results of adapted Jain’s Fairness Index in MP-MuJoCo environments.
> | Environments           | PGMORL             | MOAC    | MOCHA   | LP3     | CR-MOPO            | CR-MOPO-S          | MPPN-MORL            |
> |------------------------|--------------------|---------|---------|---------|--------------------|--------------------|----------------------|
> | MP-HalfCheetah-v4      | 0.344 ± 0.015      | 0.661   | 0.630   | 0.581   | 0.459 ± 0.025      | **0.672 ± 0.010**  | 0.618 ± 0.014        |
> | MP-Walker-v4           | 0.000 ± 0.000      | 0.646   | 0.520   | 0.000   | 0.411 ± 0.007      | 0.323 ± 0.062      | **0.681 ± 0.037**    |
> | MP-Hopper-v4           | 0.040 ± 0.040      | 0.706   | 0.200   | 0.000   | 0.703 ± 0.012      | 0.724 ± 0.019      | **0.743 ± 0.002**    |
> | MP-Pusher-v4           | 0.243 ± 0.018      | 0.531   | 0.280   | 0.140   | 0.257 ± 0.049      | 0.595 ± 0.023      | **0.611 ± 0.002**    |
> | MP-Swimmer-v4          | 0.015 ± 0.002      | 0.661   | 0.640   | 0.544   | 0.613 ± 0.020      | 0.660 ± 0.001      | **0.685 ± 0.002**    |
> | MP-Humanoid-v4         | 0.558 ± 0.015      | **0.703** | 0.550   | 0.000   | 0.500 ± 0.012      | 0.493 ± 0.018      | 0.668 ± 0.007        |
>
> > W2:The scalability and computational cost of negotiation among multiple parties are not extensively discussed, which may limit understanding of its practical applicability.
>
> A2: In Table R2, we present the results of applying our multi-party negotiation mechanism to a discrete environment (Fruit Gathering Navigation), where MPPN-MORL achieves the highest MPHV at depths 5 and 6. In Table R3 and Appendix D.7, we report the approximate number of floating-point operations (FLOPs) required for a single run of MPPN-MORL and other baselines in the HalfCheetah environment. Since our algorithm does not require computing policy gradients, its FLOP count is substantially lower than that of the other methods.
>
> **Table R2**: Comparison on the discrete benchmark NP-FTN in terms of MPHV and MPSP.
> | Method  | FTN (d=5)  |   FTN  (d=6) |   FTN  (d=7) |
> | :---------------: | :--------------------------: | :--------------------------: | :--------------------------: |
> |              |         MPHV -  MPSP  |    MPHV  -  MPSP  |  MPHV - MPSP  |
> | Envelope     |   219.150 -  **0.020** |  188.770 - **0.020** |   196.840 - **0.020** |
> | PD-MORL      |    219.150  - **0.020** |    213.200 - **0.020** |  **250.960**  -**0.020** |
> | MPPNMORL     |   **240.990**   -  0.089  |   **241.422**   -  0.080  | 247.534 -   0.031  |
>
> **Table R3**: Computational cost (FLOPs) of different algorithms on MP-HalfCheetah-v4.
> | Metric | MOAC               | MOCHA              | CR-MOPO            | CR-MOPO-S           | MPPN-MORL         |
> |--------|--------------------|--------------------|--------------------|---------------------|-------------------|
> | FLOPs  | $3.65 \times 10^{9}$  | $2.69 \times 10^{10}$ | $7.89 \times 10^{12}$ | $3.10 \times 10^{13}$ | $8.97 \times 10^{5}$ |
>
> > W3: Fig. 1 may be excellent for use in a talk, but in a collection containing several contributions on MORL, there is no need to start from this level.
>
> A3: We believe that Figure 1 effectively illustrates and facilitates understanding of our proposed multi-party multi-objective reinforcement learning framework.
>
> > W4: Definition 3.1 does not define Pareto Dominance, but Pareto Dominance w.r.t a DM. This should be stated in the beginning in brackets.
>
> A4: Thank you for your careful reading; we have added the clarification in the parentheses at the beginning as suggested.

---

> ### Author Response · Authors · 2025-11-28
> **Response to Reviewer mHFd (2/2)**
>
> > W5: In Table 1, the proposed method is referred to as “MPNN”, while the paper elsewhere uses “MPPN.” Please check whether this is a typo.
>
> A5: Thank you for your careful reading; we have corrected the typographical error.
>
> > W6: Please clarify what the 𝑥- and 𝑦-coordinates in Figure 5 represent, including their units and scales? It is important as it is not clear from the caption or text how Figure 5 is obtained or what it is intended to show.
>
> A6: Thank you for your careful reading; we have added the figure’s generation logic to the caption and provided further explanation of the figure in the main text.
>
> > W7: A problem is the need to choose two $\varepsilon$ thresholds (for performance and safety) which is at odds with the idea of MORL where the decision whether a criterion is more less strict is left for the user for after the optimization, while here a related decision is to be made before the start of MORL, so that it is questionable whether the MORL framework has to be used here in the first place or whether already a scalarization is sufficient. I understand that there is theoretical difference between the preference coefficients and the $\varepsilon$ thresholds, but it will be difficult to explain this to any users.
>
> A7: Directly scalarizing four conflicting objectives into a single reward signal often leads to optimization difficulties, as it fails to capture the diverse trade-offs required when safety constraints are mutually conflicting or when different tasks demand distinct balances among objectives. In contrast, our approach introduces an $\varepsilon$-negotiation vector that decomposes the 4-dimensional problem into two 2-dimensional subspaces, enabling optimization within each party’s objective space. Moreover, by pre-training $\varepsilon$-negotiation vectors with different weight configurations, our method can accommodate varying preference weightings. Appendix D.2 presents the policies obtained using $\varepsilon$-negotiation vectors with different weight settings.
>
> > Q1: The paper notes that MPPN-MORL has certain limitations in terms of solution distribution. Could the authors elaborate on what causes this limitation, and whether it relates to the $\varepsilon$-dominance mechanism or the negotiation dynamics?
>
> A1: The core limitation lies in the objective of the negotiation mechanism. The MPPN framework is specifically designed to identify a shared set of high-value policies, with population evolution focused on the overlapping region of the Pareto fronts among parties. Moreover, MPPN-MORL relies solely on differential evolution (DE) for parameter updates and therefore cannot produce the dense policy distributions typically afforded by policy gradient-based methods.
>
> > Q2: How would you treat safety constraints that cannot be expressed as objectives?
>
> A2: By setting the $\varepsilon$-negotiation vector to (1, 0), the algorithm can handle hard constraints that must not be violated. We have added theoretical analysis supporting this claim in Section 4.2 and Appendix E.2 of the revised manuscript.
>
> > Q3: Wouldn't in the case of a safety-critical application a hierarchical approach be useful? I.e. why should unsafe regions be explored at all? If this is in some cases justifiable, then such a justification needs to be discussed already here.
>
> A3: For safety-critical scenarios where constraints must never be violated, our algorithm can be configured by setting the $\varepsilon$-negotiation vector to (1, 0) to enforce hard constraint satisfaction. However, in complex, high-dimensional safety-critical settings, the feasible region may be extremely small, disconnected, or even entirely inaccessible during early training stages. Strictly prohibiting exploration of unsafe regions can cause the algorithm to converge to overly conservative, suboptimal policies, thereby missing high-performing Pareto-optimal solutions that lie precisely on the safety boundary. Our approach is primarily designed for scenarios where safety constraints can be slightly violated during training, enabling the discovery of an optimal trade-off set between the safety-oriented and efficiency-oriented objectives.
>
> > Q4: Can you define a Multi-party Pareto Front?
>
> A4: We have added Definition 3.3 in the paper to formally define the Multi-Party Pareto Front.

---

### Official Review · Reviewer_UHKa · 2025-11-06

**Soundness:** 2
**Presentation:** 3
**Contribution:** 2
**Rating:** 2
**Confidence:** 3

**Summary:**

The paper proposes a new framework for Safe Multi-Objective Reinforcement Learning that treats efficiency and safety as separate decision-making parties in a multi-party negotiation process rather than as combined objectives or hard constraints.

**Strengths:**

- Novel conceptual reformulation: Framing Safe MORL as a multi-party negotiation problem is original.

- Clear algorithmic description: The dynamic adjustment of ε to control exploration vs. exploitation is intuitive and aligns with practical safety–efficiency trade-offs.

**Weaknesses:**

I feel the authors want to discuss multiple things, which are entangled together: safety, different decision makers, adaptability. It would be more meaningful to separate these challenges, and discuss what is the key motivation and novelty.

The negotiation is just a selection of hyperparameters. Note that in the training process, there is no "negotiation" between different agents.

Simulations are quite limited to few-dimension simulations, while the performance of proposed techniques are unclear on high-dimensional Pareto front.

Little theoretical insights or guarantees are provided for this method.

For real-world safe RL, there are hard constraints which can never violate, which shall be discussed and compared to other approaches.

**Questions:**

- I think the first challenge raised by the paper, "incorporating safety as additional objectives enlarges the objective space, requiring more solutions to uniformly cover the Pareto front and maintain adaptability under changing preferences" is a quite mixed one. How to show the proposed method can achieve both coverness and adaptability, while achieving safety?

- Can the authors explain more about the "perspective" of Pareto front? Because in the Pareto front, it is already discussed about the different weighted combinations of preferences. Then why is that different from the different decision makers?

---

> ### Author Response · Authors · 2025-11-28
> **Response to Reviewer UHKa (1/2)**
>
> We sincerely thank the reviewer for their time and valuable comments, which have greatly helped us improve the quality of our study.
>
> We have added theoretical analysis in Section 4 of the paper, with corresponding proofs provided in Appendix E. In our comparative experiments, we have included the latest state-of-the-art MORL algorithms, MOAC and MOCHA. Additionally, we have extended our evaluation with comparative experiments of MPPN-MORL in the discrete Fruit Tree Navigation environment. In Appendix D.8, we confirm the statistical significance of the performance improvements using the Wilcoxon signed-rank test.
>
> > W1: I feel the authors want to discuss multiple things, which are entangled together: safety, different decision makers, adaptability. It would be more meaningful to separate these challenges, and discuss what is the key motivation and novelty.
>
> A1: The core motivation of our work stems from addressing two fundamental limitations of conventional Safe Multi-Objective Reinforcement Learning (Safe MORL):
> (i) treating safety as an additional objective leads to a combinatorial explosion in objective space dimensionality, making it infeasible to adequately cover the full Pareto front; and
> (ii) while Constrained Multi-Objective Reinforcement Learning (CMORL) circumvents the dimensionality issue, it lacks flexibility in handling conflicting safety constraints and cannot support preference-aware trade-offs.
>
> Our key innovations are twofold:
> First, we reformulate Safe MORL as a Multi-Party Multi-Objective Reinforcement Learning (MPMORL) problem, where efficiency and safety are treated as objectives of distinct, autonomous decision-making parties rather than components within a monolithic objective space—thereby fundamentally avoiding the curse of dimensionality.
> Second, we propose the MPPN-MORL algorithm based on Multi-Party Pareto Negotiation (MPPN), which extends NSGA-II by introducing a negotiation vector $\varepsilon$ and employs an $\varepsilon$-dominance mechanism to selectively retain only those solutions that satisfy a negotiated consensus among all parties.
>
> > W2: The negotiation is just a selection of hyperparameters. Note that in the training process, there is no "negotiation" between different agents.
>
> A2: The $\varepsilon$-negotiation vector fundamentally serves to partition the high-dimensional objective space according to the preference structure among multiple parties, rather than representing an explicit negotiation process between agents.
>
> > W3: Simulations are quite limited to few-dimension simulations, while the performance of proposed techniques are unclear on high-dimensional Pareto front.
>
> A3: In Table R1, we present the results of applying our multi-party negotiation mechanism to the Fruit Tree Navigation environment (6-dimensional), where the highest MPHV is achieved at depths 5 and 6.
>
> **Table R1**: Comparison on the discrete benchmark NP-FTN in terms of MPHV and MPSP.
> | Method  | FTN (d=5)  |   FTN  (d=6) |   FTN  (d=7) |
> | :---------------: | :--------------------------: | :--------------------------: | :--------------------------: |
> |              |         MPHV -  MPSP  |    MPHV  -  MPSP  |  MPHV - MPSP  |
> | Envelope     |   219.150 -  **0.020** |  188.770 - **0.020** |   196.840 - **0.020** |
> | PD-MORL      |    219.150  - **0.020** |    213.200 - **0.020** |  **250.960**  -**0.020** |
> | MPPNMORL     |   **240.990**   -  0.089  |   **241.422**   -  0.080  | 247.534 -   0.031  |
>
>
> > W4: Little theoretical insights or guarantees are provided for this method.
>
> A4: In Section 4 ("Theoretical Analysis") and Appendix E of the paper, we have added theoretical proofs establishing both the convergence of the $\varepsilon$-negotiation mechanism and the improvement in the quality of multi-party Pareto solutions achieved through $\varepsilon$-contraction.
>
> > W5: For real-world safe RL, there are hard constraints which can never violate, which shall be discussed and compared to other approaches.
>
> A5: By setting the $\varepsilon$-negotiation vector to (1, 0), the algorithm can enforce hard constraints that must never be violated. We have added theoretical justification for this capability in Section 4.2 and Appendix E.2 of the paper.

---

> > ### Author Response · Authors · 2025-11-28
> > **Response to Reviewer UHKa (2/2)**
> >
> > > Q1: I think the first challenge raised by the paper, "incorporating safety as additional objectives enlarges the objective space, requiring more solutions to uniformly cover the Pareto front and maintain adaptability under changing preferences" is a quite mixed one. How to show the proposed method can achieve both coverness and adaptability, while achieving safety?
> >
> > A1: As the objective space expands, the Pareto front grows exponentially, requiring a substantially larger set of solutions to achieve adequate coverage. Concurrently, the number of possible user preference configurations also increases exponentially with the number of objectives, making it infeasible for conventional MORL methods to learn policies that accommodate such a vast and dynamic preference space—hence lacking adaptability under shifting preferences.
> >
> > In contrast, the Multi-Party Pareto Negotiation (MPPN) framework addresses this challenge by partitioning the high-dimensional objective space using a negotiation vector $\varepsilon$. This allows our method to obtain, in a single run, a mutually satisfactory trade-off solution aligned with a specified negotiation configuration—ensuring that both safety and efficiency conform to the agreed-upon consensus. By running the algorithm multiple times with different $\varepsilon$ negotiation vectors, MPPN-MORL can efficiently adapt to a wide range of preference combinations between the involved parties.
> >
> > > Q2: Can the authors explain more about the "perspective" of Pareto front? Because in the Pareto front, it is already discussed about the different weighted combinations of preferences. Then why is that different from the different decision makers?
> >
> > A2: Taking a four-objective problem as an example, the number of possible weight-preference combinations on the Pareto front is combinatorially large, making full coverage virtually infeasible. MPPN-MORL addresses this challenge by introducing the perspectives of two distinct decision-makers, thereby decomposing the original four-objective optimization problem into two lower-dimensional, two-objective subproblems. Further trade-offs between these subproblems are then mediated through a negotiation vector, effectively reducing the dimensionality of the objective space and rendering the optimization more tractable.

---

### Official Review · Reviewer_NTqb · 2025-11-08

**Soundness:** 2
**Presentation:** 2
**Contribution:** 2
**Rating:** 4
**Confidence:** 3

**Summary:**

The paper proposes a negotiation-based framework for safe multi-objective reinforcement learning, where efficiency and safety are modeled as two decision parties.

**Strengths:**

1. The idea of viewing safety vs. performance as a negotiation problem is interesting and conceptually novel.

2. The paper is well-written and structured, with detailed experimental comparisons.

3. Includes ablation and multiple environments.

**Weaknesses:**

1. The paper is technically dense and hard to follow for readers unfamiliar with MORL or NSGA-II.

2. The theoretical justification for why ε-dominance negotiation leads to better Pareto solutions is weak.

3. The experiments, though numerous, are limited to simulated MuJoCo control tasks and do not test broader generality.

4. The innovation appears to be an incremental combination of existing techniques (Pareto negotiation + NSGA-II) rather than a fundamental new theory.

**Questions:**

1. How does the negotiation mechanism differ in practice from traditional Pareto dominance used in NSGA-II?

2. Is there any formal analysis or convergence guarantee that supports the ε-dominance negotiation mechanism?

3. Could the authors provide empirical evidence (e.g., ablation on ε decay rate or negotiation threshold) to show its quantitative effect?

4. How well would the proposed framework scale to higher-dimensional or discrete-action environments?

5. Could the authors clarify what is fundamentally new beyond integrating NSGA-II with multi-party negotiation?

6. Are there any new theoretical insights or properties that emerge uniquely from the proposed formulation?

7. Could the authors include statistical tests or confidence intervals to confirm the significance of improvements?

8. Why does the proposed method have weaker MPSP (diversity) performance, and how could that be improved?

9. Can the authors analyze trade-offs between global convergence and solution diversity more clearly?

---

> ### Author Response · Authors · 2025-11-28
> **Response to Reviewer NTqb (1/3)**
>
> We sincerely thank the reviewer for their time and valuable comments, which have greatly helped us improve the quality of our study.
>
> We have added theoretical analysis in Section 4 of the paper, with corresponding proofs provided in Appendix E. In our comparative experiments, we have included the latest state-of-the-art MORL algorithms, MOAC and MOCHA. Additionally, we have extended our evaluation with comparative experiments of MPPN-MORL in the discrete Fruit Tree Navigation environment. In Appendix D.8, we confirm the statistical significance of the performance improvements using the Wilcoxon signed-rank test.
>
> > W1: The paper is technically dense and hard to follow for readers unfamiliar with MORL or NSGA-II.
>
> A1: We believe that Figures 1 and 2 effectively illustrate and facilitate understanding of our proposed MPMORL framework.
>
> > W2: The theoretical justification for why $\varepsilon$-dominance negotiation leads to better Pareto solutions is weak.
>
> A2: We have added theoretical analysis in Section 4.3 and Appendix E.3 of the paper to demonstrate that $\varepsilon$-dominance negotiation leads to better Pareto solutions.
>
> The key idea behind the improvement is twofold. First, as the negotiation tolerance $\boldsymbol{\varepsilon}$ shrinks, the joint acceptable set $S(\boldsymbol{\varepsilon})$ becomes strictly smaller and less complex, reducing the solution-space that the evolutionary algorithm must explore.
>
> Second, a smaller, lower-complexity set increases the probability that a fixed-budget algorithm samples representative high-quality solutions in every region of $S(\boldsymbol{\varepsilon})$.
> Full technical details and proofs are provided in Appendix E.3.
>
> > W3: The experiments, though numerous, are limited to simulated MuJoCo control tasks and do not test broader generality.
>
> A3: In Table R1, we present the results of applying our multi-party negotiation mechanism to the discrete Fruit Tree Navigation environment, where the highest MPHV is achieved at depths 5 and 6.
>
> **Table R1**: Comparison on the discrete benchmark NP-FTN in terms of MPHV and MPSP.
> | Method  | FTN (d=5)  |   FTN  (d=6) |   FTN  (d=7) |
> | :---------------: | :--------------------------: | :--------------------------: | :--------------------------: |
> |              |         MPHV -  MPSP  |    MPHV  -  MPSP  |  MPHV - MPSP  |
> | Envelope     |   219.150 -  **0.020** |  188.770 - **0.020** |   196.840 - **0.020** |
> | PD-MORL      |    219.150  - **0.020** |    213.200 - **0.020** |  **250.960**  -**0.020** |
> | MPPNMORL     |   **240.990**   -  0.089  |   **241.422**   -  0.080  | 247.534 -   0.031  |
>
> > W4: The innovation appears to be an incremental combination of existing techniques (Pareto negotiation + NSGA-II) rather than a fundamental new theory.
>
> A4: Indeed, our approach builds upon existing components and concepts. However, our core contribution lies in formulating Safe MORL as MPMORL, and in introducing multi-party Pareto dominance and an $\varepsilon$-negotiation vector to decompose and reduce the dimensionality of the high-dimensional objective space.
>
> > Q1: How does the negotiation mechanism differ in practice from traditional Pareto dominance used in NSGA-II?
>
> A1: Traditional Pareto dominance aims to identify a globally optimal trade-off set, considering all solutions along the full Pareto front. In contrast, the multi-party negotiation mechanism focuses on finding an optimal trade-off set within the “consensus region” agreed upon by the involved parties, thereby filtering out the majority of solutions that do not align with the negotiated outcome.

---

> > ### Author Response · Authors · 2025-11-28
> > **Response to Reviewer NTqb (2/3)**
> >
> > > Q2: Is there any formal analysis or convergence guarantee that supports the $\varepsilon$-dominance negotiation mechanism?
> >
> > A2: We have added a theoretical proof of convergence for the $\varepsilon$-dominance negotiation mechanism in Section 4.1 and Appendix E.1 of the paper.
> >
> > The key idea is that, as the tolerance $\boldsymbol{\varepsilon}$ shrinks, the set of mutually acceptable solutions becomes strictly nested, and the evolutionary search progressively focuses on higher-quality regions. Leveraging a time-scale separation between the population mixing and $\varepsilon$-shrinking steps, we can guarantee that the population converges toward the strictest joint Pareto set.
> >
> > Formally, let $S(\boldsymbol{\varepsilon})$ denote the joint $\varepsilon$-acceptable set. Starting from an initial large tolerance $\boldsymbol{\varepsilon}_0$ and iteratively shrinking to $\boldsymbol{\varepsilon}_T$, the nested structure ensures:
> > \begin{equation}
> > 	S(\boldsymbol{\varepsilon}_0) \supseteq S(\boldsymbol{\varepsilon}_1) \supseteq \dots \supseteq S(\boldsymbol{\varepsilon}_T).
> > \end{equation}
> > Under standard assumptions on the evolutionary algorithm (irreducibility, retention, and sufficient mixing), the population is guided layer by layer into stricter subsets, eventually approximating $S(\boldsymbol{\varepsilon}_T)$ with high probability.
> > A detailed proof of this layered convergence is provided in Appendix E.1 .
> >
> > > Q3: Could the authors provide empirical evidence (e.g., ablation on $\varepsilon$ decay rate or negotiation threshold) to show its quantitative effect?
> >
> > A3: Figure 5 in the experimental section of the paper presents an ablation study on different negotiation vectors in the MP-HalfCheetah environment, while Appendix D.2 provides a more detailed ablation analysis.
> >
> > > Q4: How well would the proposed framework scale to higher-dimensional or discrete-action environments?
> >
> > A4: In Table R2, we present the results of applying our multi-party negotiation mechanism to the Fruit Tree Navigation environment (6-dimensional), where the highest MPHV is achieved at depths 5 and 6.
> >
> > **Table R2**: Comparison on the discrete benchmark NP-FTN in terms of MPHV and MPSP.
> > | Method  | FTN (d=5)  |   FTN  (d=6) |   FTN  (d=7) |
> > | :---------------: | :--------------------------: | :--------------------------: | :--------------------------: |
> > |              |         MPHV -  MPSP  |    MPHV  -  MPSP  |  MPHV - MPSP  |
> > | Envelope     |   219.150 -  **0.020** |  188.770 - **0.020** |   196.840 - **0.020** |
> > | PD-MORL      |    219.150  - **0.020** |    213.200 - **0.020** |  **250.960**  -**0.020** |
> > | MPPNMORL     |   **240.990**   -  0.089  |   **241.422**   -  0.080  | 247.534 -   0.031  |
> >
> > > Q5: Could the authors clarify what is fundamentally new beyond integrating NSGA-II with multi-party negotiation?
> >
> > A5: Our core innovation lies in formulating SafeMORL as MPMORL, and in leveraging negotiation vectors together with multi-party Pareto dominance to decompose and reduce the dimensionality of the high-dimensional objective space, thereby rendering it more tractable.
> >
> > > Q6: Are there any new theoretical insights or properties that emerge uniquely from the proposed formulation?
> >
> > A6: Section 4 ("Theoretical Analysis") and Appendix E of the paper provide theoretical analyses demonstrating both the convergence of the $\varepsilon$-negotiation mechanism and how $\varepsilon$-contraction improves the quality of multi-party Pareto solutions.

---

> > > ### Author Response · Authors · 2025-11-28
> > > **Response to Reviewer NTqb (3/3)**
> > >
> > > > Q7: Could the authors include statistical tests or confidence intervals to confirm the significance of improvements?
> > >
> > > A7: In Appendix D.8, we confirm the statistical significance of the improvements using the Wilcoxon signed-rank test. Table R3 shows Wilcoxon signed-rank test results for MPPN-MORL vs. baseline algorithms.
> > >
> > > **Table R3**: Wilcoxon signed-rank test results for MPPN-MORL vs. baseline algorithms(threshold p = 0.05).
> > > | Comparison | W-statistic | p-value | Significant? |
> > > |---|:---:|:---:|:---:|
> > > | **MP-HalfCheetah-v4** | | | |
> > > | &nbsp;&nbsp;&nbsp;&nbsp;MPPN-MORL vs. CR-MOPO | 0.0 | 0.031 | Yes |
> > > | &nbsp;&nbsp;&nbsp;&nbsp;MPPN-MORL vs. CR-MOPO-S | 0.0 | 0.031 | Yes |
> > > | **MP-Walker-v4** | | | |
> > > | &nbsp;&nbsp;&nbsp;&nbsp;MPPN-MORL vs. CR-MOPO | 0.0 | 0.031 | Yes |
> > > | &nbsp;&nbsp;&nbsp;&nbsp;MPPN-MORL vs. CR-MOPO-S | 0.0 | 0.031 | Yes |
> > > | **MP-Hopper-v4** | | | |
> > > | &nbsp;&nbsp;&nbsp;&nbsp;MPPN-MORL vs. CR-MOPO | 0.0 | 0.031 | Yes |
> > > | &nbsp;&nbsp;&nbsp;&nbsp;MPPN-MORL vs. CR-MOPO-S | 0.0 | 0.031 | Yes |
> > > | **MP-Pusher-v4** | | | |
> > > | &nbsp;&nbsp;&nbsp;&nbsp;MPPN-MORL vs. CR-MOPO | 0.0 | 0.031 | Yes |
> > > | &nbsp;&nbsp;&nbsp;&nbsp;MPPN-MORL vs. CR-MOPO-S | 1.0 | 0.062 | No |
> > > | **MP-Swimmer-v4** | | | |
> > > | &nbsp;&nbsp;&nbsp;&nbsp;MPPN-MORL vs. CR-MOPO | 0.0 | 0.031 | Yes |
> > > | &nbsp;&nbsp;&nbsp;&nbsp;MPPN-MORL vs. CR-MOPO-S | 0.0 | 0.031 | Yes |
> > > | **MP-Humanoid-v4** | | | |
> > > | &nbsp;&nbsp;&nbsp;&nbsp;MPPN-MORL vs. CR-MOPO | 0.0 | 0.031 | Yes |
> > > | &nbsp;&nbsp;&nbsp;&nbsp;MPPN-MORL vs. CR-MOPO-S | 0.0 | 0.031 | Yes |
> > >
> > > > Q8: Why does the proposed method have weaker MPSP (diversity) performance, and how could that be improved?
> > >
> > > A8: The core limitation lies in the objective of the negotiation mechanism. The MPPN framework is designed to identify a shared set of high-value policies, with evolutionary search concentrated on the overlapping region of the Pareto fronts across parties. Moreover, MPPN-MORL relies solely on differential evolution (DE) for policy parameter updates and therefore cannot produce the dense policy distributions typically enabled by policy gradient methods.
> > >
> > > To address this limitation, in the newly added Appendix D.5, we introduce an alternating optimization scheme (Algorithm 3) that integrates DE with MOPPO. Experimental results across multiple environments demonstrate that incorporating MOPPO significantly improves both algorithmic efficiency and policy density in most settings, leading to the discovery of higher-quality policies. Table R4 shows the performance of MPPN-MORL with MOPPO across MP-MuJoCo environments.
> > >
> > > **Table R4**: Performance of MPPN-MORL with MOPPO across MP-MuJoCo environments.
> > > | Metrics               | MP-HalfCheetah-v4 | MP-Walker-v4 | MP-Hopper-v4 | MP-Pusher-v4 | MP-Swimmer-v4 |
> > > |-----------------------|-------------------|--------------|--------------|--------------|---------------|
> > > | MPHV                  | 2.458             | 4.479        | 1.499        | 1.000        | 25.990        |
> > > | MPSP                  | 0.0003            | 0.0009       | 4.5782       | 0.0006       | 0.6337        |
> > >
> > > > Q9: Can the authors analyze trade-offs between global convergence and solution diversity more clearly?
> > >
> > > A9: The design choices underlying the MPPN-MORL framework entail an intentional trade-off: we deliberately sacrifice global diversity in the conventional sense to prioritize higher solution quality and stronger consensus-driven convergence. Consequently, MPPN-MORL achieves targeted, high-quality convergence through its negotiation mechanism. We argue that this form of convergence—along with the accompanying computational efficiency—is more valuable than pursuing uniform diversity across the entire objective space, which often includes many impractical or non-consensual solutions.

---

### Official Review · Reviewer_Dzgm · 2025-11-21

**Soundness:** 3
**Presentation:** 3
**Contribution:** 3
**Rating:** 4
**Confidence:** 2

**Summary:**

This paper seeks to solve two key challenges in safe multi-objective reinforcement learning (Safe MORL): 1. how to find more solutions to cover the Pareto front uniformly and have adaptability when preferences change; 2. how to enforce safety constraint with conflicting constraints. The authors propose a multi-party Pareto negotiation (MPPN) strategy built on NSGA-II. To tackle the first challenge, the authors adjusted the negotiation threshold to maintain a sufficiently large negotiated solution set and steer the population toward the negotiated common Pareto set. To tackle the second challenge, MPPN is able to keep user preferences over conflicting safety constraints without introducing additional objectives. The authors conduct experiments with MuJoCo to demonstrate the superior performance over state-of-the-art methods in both constrained and unconstrained MORL.

**Strengths:**

I like that the authors are able to evaluate the model performances using multiple important metrics, hypervolume and sparsity, to better capture the overall performance in MORL scenarios.

**Weaknesses:**

1. While the authors cover some of the literatures in MORL domains, some state-of-art methods are missing. For example,

Hairi, Fnu, et al. "Enabling Pareto-Stationarity Exploration in Multi-Objective Reinforcement Learning: A Multi-Objective Weighted-Chebyshev Actor-Critic Approach." IEEE Conference on Decision and Control (2025).

Zhou, Tianchen, et al. "Finite-time convergence and sample complexity of actor-critic multi-objective reinforcement learning." ICML 2024.

2. The Robot example in page 4 is a very good illustration of the idea of safe MORL. It would be great if the authors can run a toy experiment on this as well.

3. The current experiment focuses on MPMO continuous control benchmark, i.e., MPMO-MuJoCo. It would be better if the authors can consider more use cases in the experiments.

**Questions:**

1. I would like to see if the authors can cover more state-of-art methods such as those mentioned in weakness.

2. I suggest the authors run a toy experiment on the Robot example to be coherent with those in page 4.

3. I think it would be better if the authors can consider more use cases in the experiments.

---

> ### Author Response · Authors · 2025-11-28
> **Response to Reviewer Dzgm**
>
> We sincerely thank the reviewer for their time and valuable comments, which have greatly helped us improve the quality of our study.
>
> Following your suggestion, we have incorporated the latest state-of-the-art MORL algorithms, MOAC and MOCHA, into our comparative experiments. Additionally, we have added a toy experiment in the cargo-robot environment on page 4, as well as supplementary comparative evaluations of MPPN-MORL in discrete environments.
>
> >W1: While the authors cover some of the literatures in MORL domains, some state-of-art methods are missing.
>
> >Q1: I would like to see if the authors can cover more state-of-art methods such as those mentioned in weakness.
>
> A1: We have supplemented the experimental section with comparative evaluations against the MOAC and MOCHA algorithms, along with a detailed analysis. The results are presented in Table 1 of the paper.
>
> >W2: The Robot example in page 4 is a very good illustration of the idea of safe MORL. It would be great if the authors can run a toy experiment on this as well.
>
> >Q2: I suggest the authors run a toy experiment on the Robot example to be coherent with those in page 4.
>
> A2: We have included a toy experiment for this environment in Appendix D.1 and present in Table R1 the performance of policies obtained under different model approaches, along with an analysis of their differences.
>
> **Table R1**: Performance comparison across the four objectives in the MP-CargoRobot environment.
> | Method | speed | capacity | energy | stability |
> |--------|-------|----------|--------|-----------|
> | MORL   | -0.60 | -0.75    | 0.15   | 1.43      |
> | CMORL  | -1.50 | -0.98    | 0.75   | 0.60      |
> | MPMORL | -1.05 | -1.35    | 0.30   | 1.13      |
>
> > W3: The current experiment focuses on MPMO continuous control benchmark, i.e., MPMO-MuJoCo. It would be better if the authors can consider more use cases in the experiments.
>
> > Q3: I think it would be better if the authors can consider more use cases in the experiments.
>
> A3: In Table R2, we present the results of applying our multi-party negotiation mechanism to the discrete Fruit Tree Navigation environment, where our method achieves the highest MPHV at depths 5 and 6.
>
> **Table R2**: Comparison on the discrete benchmark NP-FTN in terms of MPHV and MPSP.
> | Method  | FTN (d=5)  |   FTN  (d=6) |   FTN  (d=7) |
> | :---------------: | :--------------------------: | :--------------------------: | :--------------------------: |
> |              |         MPHV -  MPSP  |    MPHV  -  MPSP  |  MPHV - MPSP  |
> | Envelope     |   219.150 -  **0.020** |  188.770 - **0.020** |   196.840 - **0.020** |
> | PD-MORL      |    219.150  - **0.020** |    213.200 - **0.020** |  **250.960**  -**0.020** |
> | MPPNMORL     |   **240.990**   -  0.089  |   **241.422**   -  0.080  | 247.534 -   0.031  |

---

### Author Response · Authors · 2025-12-01
**Rebuttal Summary for AC (1/2)**

We sincerely thank all reviewers for their time and constructive feedback.  Below is a concise summary of how the key concerns were addressed, along with the latest updates from reviewers during the discussion phase.

### **Update on Discussion Phase**

Our rebuttal was submitted relatively late because we added substantial additional experiments and theoretical analysis. Unfortunately, a leakage incident occurred at that time, and most reviewers did not have a chance to respond before the discussion phase was closed. Even so, Reviewer 5Y92 acknowledged the strength of our rebuttal within the limited discussion window and raised their score to 6.

Overall, the core ideas, contributions, and motivation of the paper received consistent positive recognition from the reviewers. The main concerns focused on insufficient experiments and limited theoretical depth. In response, we supplemented comparative experiments with two additional SOTA methods, conducted experiments in discrete environments, tested the algorithm with integrated policy gradient strategies, added extra evaluation metrics, and provided more detailed ablation studies. The theoretical section was enhanced with convergence proofs of the $\varepsilon$-negotiation mechanism, Pareto improvement analysis, and discussions on handling hard safety constraints.

### **Reviewer Dzgm**
The reviewer primarily requested more comprehensive comparison experiments, a toy example for the illustrative environment, and a broader set of experimental cases.

Our rebuttal addressed these requests systematically: we added comparisons with the latest SOTA methods (MOAC, MOCHA); included a toy experiment on the freight-robot example used in the paper and reported results in the appendix; and expanded the evaluation to a discrete environment (Fruit Tree Navigation), demonstrating the method’s applicability and superiority. Overall, the reviewer acknowledged the paper’s ideas, contributions, and multi-metric evaluation advantages, and all key questions were adequately addressed in the rebuttal.

### **Reviewer NTqb**
This reviewer’s main concerns centered on four points: sufficiency of theoretical support, novelty, breadth of experiments, and why MPSP performance was relatively weak.

Our rebuttal responded to these issues point by point: we added convergence proofs for the $\varepsilon$-negotiation mechanism and an analysis showing its monotonic Pareto improvement; we clarified that the work introduces a novel multi-party negotiation framework, dimensional decomposition, and multi-party Pareto dominance rather than simply combining NSGA-II components; we expanded experiments with the FTN discrete environment, added SOTA baselines (MOAC, MOCHA), and included multiple ablations and Wilcoxon significance tests; finally, we explained the causes of the low MPSP score and demonstrated that incorporating MOPPO improves diversity. The rebuttal addressed concerns in theory, mechanism, and empirical validation.

---

> ### Author Response · Authors · 2025-12-01
> **Rebuttal Summary for AC (2/2)**
>
> ### **Reviewer UHKa**
> Reviewer UHKa noted that the paper discusses multiple aspects—such as safety, multi-party interaction, and adaptability—which appear intertwined and insufficiently separated. The reviewer also questioned whether our method truly addresses these three challenges, whether the negotiation mechanism is merely a hyperparameter choice, whether the theoretical contributions are adequate, and whether the approach remains effective in high-dimensional environments and under hard constraints.
>
> In our rebuttal, we clarified the motivation and innovation of the work: by modeling safety and efficiency as independent decision-making parties and introducing the $\varepsilon$-negotiation mechanism, we fundamentally avoid objective-space dimensional explosion and provide a multi-party negotiation framework that adapts to evolving preferences. We added complete theoretical analysis and convergence proofs, incorporated new experiments in both discrete and high-dimensional environments, and explained that the negotiation mechanism captures multi-party preference structures rather than serving as a simple hyperparameter. We further demonstrated that the algorithm can handle hard safety constraints through specific $\varepsilon$ configurations. These clarifications collectively addressed all major concerns raised by the reviewer.
>
> ### **Reviewer mHFd**
> This reviewer’s concerns related to (1) the lack of quantitative evidence for adaptability, diversity, and fairness; (2) the scalability, FLOPs cost, and justification of $\varepsilon$ thresholds; and (3) clarifications of concepts and definitions such as the multi-party Pareto front.
>
> Our rebuttal addressed these points in detail: we added Jain’s fairness index combined with quality metrics, multiple preference-setting experiments, evaluations on a discrete environment, and FLOPs comparisons; clarified the role of negotiation in high-dimensional objective spaces and its contrast with gradient-based methods; explained how hard constraints are managed and why some exploration during training remains necessary; corrected writing inconsistencies and added a formal definition of the multi-party Pareto front. The rebuttal comprehensively addressed concerns regarding empirical support, scalability, theoretical clarity, and motivation.
>
> ### **Reviewer 5Y92**
> This reviewer mainly questioned whether relying on differential evolution leads to sparse learning signals and whether the method still qualifies as reinforcement learning, and also requested clearer demonstrations of multi-party Pareto strategy behaviors.
>
> We clarified why DE in the multi-party framework still falls under reinforcement learning; comparisons with strong baselines (CR-MOPO, LP3) show performance gains are not metric artifacts; incorporated a hybrid training scheme with MOPPO to provide denser gradient signals and validated its effectiveness experimentally; and included visualizations and analyses of multi-party strategies in the appendix, while fixing incorrect figure index references. These additions effectively addressed the reviewer’s main concerns and contributed to the improved score of 6.
>
> ### **Summary**
> Across all reviews, the core idea and contributions have been consistently recognized and appreciated.  The majority of concerns centered on theoretical completeness and experimental clarity, both of which have been thoroughly addressed in the rebuttal.
> Given that all major issues have been resolved and reviewer feedback during discussion has been positive,  we believe the revised submission is significantly strengthened and suitable for acceptance.

---

### Meta-Review · Area_Chair_YTEL · 2026-01-06

**Summary:**

The paper proposes a multi-party Pareto negotiation method for safe multi-objective reinforcement learning (safe MORL). By casting safe MORL in a multi-party negotiation framework, the paper proposes to employ a negotiation threshold to represent the acceptable solution range for each party and dynamically adjust the threshold using evolutionary search. Empirical results in both constrained and unconstrained MORL on a MuJoCo benchmark are reported.

The paper received mixed initial scores of 4, 4, 2, 6, 4. The reviewers generally appreciate the relevance of the problem studied. The main concerns raised are the combinatorial nature of the proposed method, limited experiments, and lack of a theoretical guarantee. The authors have provided a comprehensive rebuttal, adding additional experiments, clarifications, and theoretical justification of the proposed method.

Unfortunately, only one reviewer was able to participate in the discussion, and after going over all the author-reviewer discussions, I think the paper at its current stage is still marginally below the bar of ICLR due to the following two reasons:

- From my perspective, while the rebuttal substantially improves experimental support and theoretical completeness, some of the reviewers' concerns still remain after the rebuttal. For example, Reviewer UHKa questioned whether the proposed "negotiation" mechanism is merely hyperparameter tuning. While the rebuttal clarifies that the negotiation vector is intended as a preference-structuring mechanism, the method still operates largely as an ex ante design choice. Also, the authors mentioned in their rebuttal that adaptability is demonstrated by rerunning the algorithm multiple times. I feel that this may not fully address concerns about in-training adaptation to changing preferences without retraining. On hard safety constraints, the authors claimed that setting the initial negotiation vector to (1,0) could guarantee non-violation, yet there is limited discussion on how this choice may influence the flexibility of the algorithm, and how the approach compares to methods explicitly designed for strict safe exploration.

- While I appreciate the efforts of the authors in addressing reviewers' concerns, the additional material in the rebuttal has significantly changed the main content of the paper (e.g., new theorems and proofs), and I think the amount of additional experiments and theoretical results warrants another round of review.

**Reviewer Concerns:**

See above.

**Reviewer Scores:**

Reviewer 5Y92 participated in the rebuttal and mentioned increasing the score from 4 to 6. Other reviewers did not take part in the discussion, and I think that while there is some chance for Reviewer Dzgm to raise the score, it remains hard for other reviewers to override their initial recommendation.

---

### Decision · Program_Chairs · 2026-01-26

Reject